# Nonparametric Data Attribution for Diffusion Models

Yutian Zhao [* 1 2]   Chao Du [* 1]   Xiaosen Zheng [1 3]   Tianyu Pang [1]   Min Lin [1]

## Abstract

Data attribution for generative models seeks to quantify the influence of individual training examples on model outputs. Existing methods for diffusion models typically require access to model gradients or retraining, limiting their applicability in proprietary or large-scale settings. We propose a *nonparametric* attribution method that operates directly on data, measuring influence via patch-level similarity between generated and training images. Our approach is grounded in the analytical form of the optimal score function, extends naturally to multiscale representations, and remains computationally efficient through convolution-based acceleration. In addition to producing spatially interpretable attributions, our framework uncovers patch-level correspondences between training data and generated outputs. Experiments show that NDA substantially outperforms existing nonparametric baselines and achieves the best attribution performance among the evaluated methods that do not use target-model parameters or gradients.

## 1. Introduction

Recent advances in generative models, particularly diffusion models (Ho et al., 2020; Song et al., 2021), have led to significant progress in image synthesis and editing (Rombach et al., 2022; Ramesh et al., 2022; Meng et al., 2022). As these powerful models are trained on increasingly large-scale datasets that often contain private, copyrighted, or low-quality content, concerns around data transparency, accountability, and ethical use are growing (Carlini et al., 2023; Saveri & Butterick, 2023). These concerns motivate the study of *data attribution*: identifying the influence of individual training examples on a given generation. Ef-

fective data attribution not only supports the responsible deployment of generative models but also enables various downstream applications, including interpreting model behavior (Koh & Liang, 2017; Sui et al., 2021; Ilyas et al., 2022), detecting mislabeled or poisoned data (Jia et al., 2021), guiding data valuation (Nohyun et al., 2023), and improving dataset quality through informed curation (Khanna et al., 2018; Jia et al., 2021; Liu et al., 2021).

Much progress has been made in attributing image generations to training data. Retraining-based methods (Ghorbani & Zou, 2019; Ilyas et al., 2022) assess how generations change when specific training data are removed. While effective, these methods typically require retraining the model tens of thousands of times on different data subsets (Ghorbani & Zou, 2019), making them computationally expensive. To improve efficiency, recent works (Zheng et al., 2024; Lin et al., 2025; Mlodozeniec et al., 2025) adopt approximations based on additive attribution scores (Park et al., 2023a), enabling scalable attribution on large datasets. A common assumption in these methods is access to model gradients, i.e., full access to the generative model. However, this assumption is not always practical. For example, in scenarios where users seek copyright protection against infringement by proprietary models (Somepalli et al., 2023; Zhao et al., 2023), the model gradients may not be accessible. Furthermore, when attribution is intended to support tasks such as data selection (Gu et al., 2025b), training a generative model solely for attribution may be prohibitively costly or infeasible.

These challenges call for an attribution method that does not require access to target-model parameters or gradients, either to attribute outputs from proprietary or otherwise restricted-access generative models or to serve as a practical surrogate without the cost of training a separate generative model solely for attribution. Existing methods (Zheng et al., 2024; Mlodozeniec et al., 2025) typically measure the similarity between generated images and training data in feature spaces, but often perform poorly as they disregard the behavior of the generative model. Effective attribution for diffusion models in this restricted-access setting remains an open challenge.

In this paper, we present a *nonparametric* approach to image data attribution in diffusion models, which directly quantifies the influence of training samples on generated outputs

---

[*]Equal contribution   [1]Sea AI Lab, Singapore   [2]Department of Mathematics, National University of Singapore, Singapore   [3]Singapore Management University, Singapore. Correspondence to: Chao Du <duchao@sea.com>.

*Proceedings of the 43$^{rd}$ International Conference on Machine Learning*, Seoul, South Korea. PMLR 306, 2026. Copyright 2026 by the author(s).

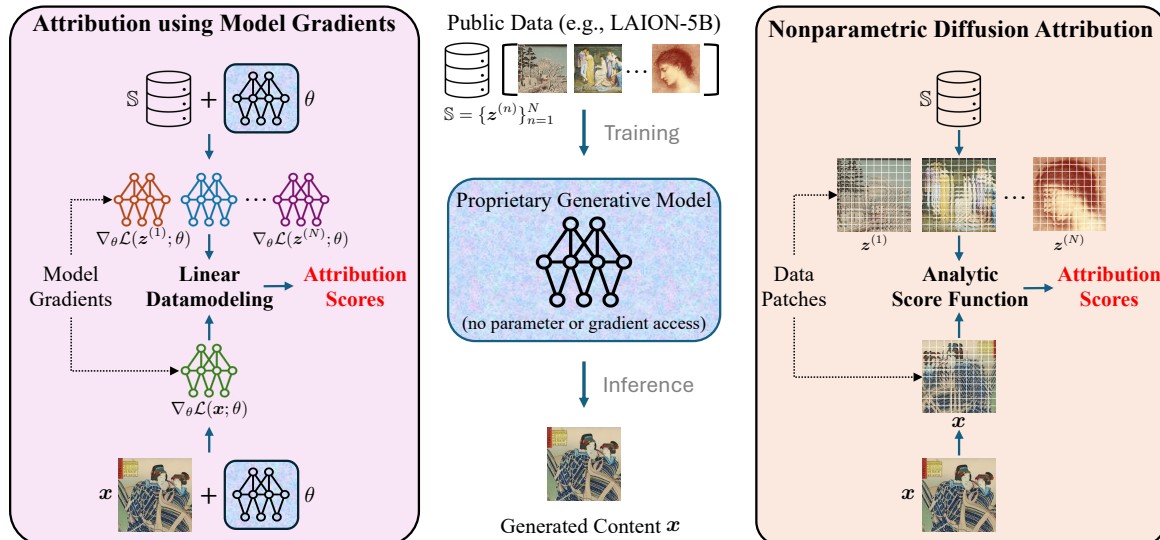

*Figure 1.* Schematic illustration of attribution methods. **Left:** Model-based attribution relies on gradients and requires parameter access. **Right:** Our *Nonparametric Data Attribution* (**NDA**) compares local image patches via an analytic score function, enabling attribution without access to target-model parameters or gradients.

through patch-level comparisons. Drawing inspiration from analytical expressions of the score function in diffusion models (Gu et al., 2025a; Kamb & Ganguli, 2025), we extract local patches from both generated images and training samples and compute attribution scores based on pairwise distances, as illustrated in Fig. 1. Unlike prior methods that require retraining or gradient access, NDA computes attributions directly from a generated image and the available training data using a chosen forward diffusion process, without requiring knowledge of the target model's architecture or training objective, or access to its parameters or gradients. Moreover, it provides fine-grained, patch-level attribution, enabling the localization of influential training regions and offering interpretable insights into generation behavior.

## 2. Preliminaries

Our approach is grounded in the optimal score function of diffusion models. In this section, we begin with a brief overview of diffusion models and their optimal empirical score functions, followed by a review of data attribution techniques and evaluation protocols.

### 2.1. Diffusion Models

Diffusion models (Ho et al., 2020; Song et al., 2021) are a class of probabilistic generative models that learn to approximate a data distribution $q(\boldsymbol{x})$ by modeling a parametrized Markovian process $p_\theta(\boldsymbol{x})$. Specifically, a diffusion model defines a forward process that transforms a clean data sample $\boldsymbol{x} \sim q(\boldsymbol{x})$ into a noisy sequence $\boldsymbol{x}_{1:T} = \boldsymbol{x}_1, \cdots, \boldsymbol{x}_T$ by gradually adding Gaussian noise. The transition probability is given by $q(\boldsymbol{x}_t|\boldsymbol{x}_{t-1}) \triangleq \mathcal{N}(\boldsymbol{x}_t|\sqrt{1-\beta_t}\boldsymbol{x}_{t-1}, \beta_t\mathbf{I})$, where $\{\beta_t\}_{t=1}^T$ denotes a predefined variance schedule. A

key property of the forward process is that $\boldsymbol{x}_t$ can be sampled in closed form at any timestep $t$:

$$q(\boldsymbol{x}_t|\boldsymbol{x}) = \mathcal{N}(\boldsymbol{x}_t|\sqrt{\bar{\alpha}_t}\boldsymbol{x}, (1-\bar{\alpha}_t)\mathbf{I}), \tag{1}$$

where $\bar{\alpha}_t \triangleq \prod_{s=1}^t \alpha_s$ and $\alpha_t \triangleq 1 - \beta_t$. As $t$ increases from 0 to $T$, the noise level increases and $\bar{\alpha}_t$ decays from 1 to 0. Consequently, the marginal distribution $q_t(\boldsymbol{x}_t)$ approaches a standard Gaussian, i.e., $q_T(\boldsymbol{x}_T) \approx \mathcal{N}(0, \mathbf{I})$. The reverse process is learned by approximating the reverse conditionals $q(\boldsymbol{x}_{t-1}|\boldsymbol{x}_t)$ using a neural network model $p_\theta(\boldsymbol{x}_{t-1}|\boldsymbol{x}_t) \triangleq \mathcal{N}(\boldsymbol{x}_{t-1}|\boldsymbol{\mu}_\theta(\boldsymbol{x}_t, t), \sigma_t^2\mathbf{I})$, where the variances $\{\sigma_t^2\}$ are typically chosen as hyperparameters (Bao et al., 2022). In practice, diffusion models are usually trained to predict the added noise $\boldsymbol{\epsilon}_\theta(\boldsymbol{x}_t, t)$, which relates to the mean via $\boldsymbol{\mu}_\theta(\boldsymbol{x}_t, t) = \frac{1}{\sqrt{\alpha_t}}(\boldsymbol{x}_t - \frac{\beta_t}{\sqrt{1-\bar{\alpha}_t}}\boldsymbol{\epsilon}_\theta(\boldsymbol{x}_t, t))$ (Ho et al., 2020). The model is trained by minimizing a variational bound on the negative log-likelihood:

$$\mathcal{L}_{\text{ELBO}}(\boldsymbol{x}; \theta) = \mathbb{E}_{\boldsymbol{\epsilon}, t}\left[w_t \|\boldsymbol{\epsilon}_\theta(\boldsymbol{x}_t, t) - \boldsymbol{\epsilon}\|_2^2\right], \tag{2}$$

where $\boldsymbol{\epsilon} \sim \mathcal{N}(\mathbf{0}, \mathbf{I})$ and $w_t \triangleq \frac{\beta_t^2}{2\sigma_t^2\alpha_t(1-\bar{\alpha}_t)}$. Let the training dataset be $\mathbb{S} \triangleq \{\boldsymbol{z}^{(n)}|\boldsymbol{z}^{(n)} \sim q\}_{n=1}^N$. The empirical training objective is then $\mathcal{L}_{\text{ELBO}}(\mathbb{S}; \theta) = \frac{1}{N}\sum_{n=1}^N \mathcal{L}_{\text{ELBO}}(\boldsymbol{z}^{(n)}; \theta)$. To enhance sample quality, a simplified training objective is often used (Ho et al., 2020):

$$\mathcal{L}_{\text{Simple}}(\boldsymbol{x}; \theta) = \mathbb{E}_{\boldsymbol{\epsilon}, t}\left[\|\boldsymbol{\epsilon}_\theta(\boldsymbol{x}_t, t) - \boldsymbol{\epsilon}\|^2\right], \tag{3}$$

with the empirical objective on $\mathbb{S}$ given by $\mathcal{L}_{\text{Simple}}(\mathbb{S}; \theta) = \frac{1}{N}\sum_{n=1}^N \mathcal{L}_{\text{Simple}}(\boldsymbol{z}^{(n)}; \theta)$.

## 2.2. Optimal Score Functions

A key property ([Song et al., 2021](#); [Kingma & Gao, 2023](#)) of diffusion models is that the optimal noise prediction function $\boldsymbol{\epsilon}^*(\boldsymbol{x}_t, t)$ is closely related to the *score function* $\boldsymbol{s}(\boldsymbol{x}_t, t) \triangleq \nabla_{\boldsymbol{x}_t} \log q_t(\boldsymbol{x}_t)$ via:

$$\boldsymbol{s}(\boldsymbol{x}_t, t) = -\frac{\boldsymbol{\epsilon}^*(\boldsymbol{x}_t, t)}{\sqrt{1 - \bar{\alpha}_t}}. \tag{4}$$

Intuitively, for a finite dataset $\mathbb{S} = \{\boldsymbol{z}^{(n)}\}_{n=1}^N$, the marginal distribution $q_t(\boldsymbol{x}_t)$ at time $t$ becomes a Gaussian mixture centered at the scaled data points $\sqrt{\bar{\alpha}_t} \boldsymbol{z}^{(n)}$: $q_t(\boldsymbol{x}_t) = \frac{1}{N} \sum_{n=1}^N \mathcal{N}\left(\boldsymbol{x}_t | \sqrt{\bar{\alpha}_t} \boldsymbol{z}^{(n)}, (1 - \bar{\alpha}_t)\mathbf{I}\right)$. The score function $\boldsymbol{s}(\boldsymbol{x}_t, t)$ therefore admits an analytical form:

$$\boldsymbol{s}(\boldsymbol{x}_t, t) = \frac{1}{1 - \bar{\alpha}_t} \sum_{n=1}^N (\sqrt{\bar{\alpha}_t} \boldsymbol{z}^{(n)} - \boldsymbol{x}_t) W_t(\boldsymbol{z}^{(n)} | \boldsymbol{x}_t), \tag{5}$$

where $W_t(\boldsymbol{z}^{(n)} | \boldsymbol{x}_t)$ is a weighting term defined as:

$$W_t(\boldsymbol{z}^{(n)} | \boldsymbol{x}_t) = \frac{\mathcal{N}\left(\boldsymbol{x}_t | \sqrt{\bar{\alpha}_t} \boldsymbol{z}^{(n)}, (1 - \bar{\alpha}_t)\mathbf{I}\right)}{\sum_{n'=1}^N \mathcal{N}\left(\boldsymbol{x}_t | \sqrt{\bar{\alpha}_t} \boldsymbol{z}^{(n')}, (1 - \bar{\alpha}_t)\mathbf{I}\right)}. \tag{6}$$

The score $\boldsymbol{s}(\boldsymbol{x}_t, t)$ can therefore be interpreted as a posterior-weighted average of the vectors pointing from $\boldsymbol{x}_t$ to the scaled training examples $\sqrt{\bar{\alpha}_t} \boldsymbol{z}^{(n)}$, scaled by $(1 - \bar{\alpha}_t)^{-1}$. Here, $W_t(\boldsymbol{z}^{(n)} | \boldsymbol{x}_t)$ denotes the posterior probability that $\boldsymbol{x}_t$ was transformed from $\boldsymbol{z}^{(n)}$ at $t=0$ under the forward process.

## 2.3. Data Attribution and Evaluation Metrics

Given a training dataset $\mathbb{S}$ and a generated sample $\boldsymbol{x}$, the goal of data attribution is to quantify the influence of each training example in $\mathbb{S}$ on the generation of $\boldsymbol{x}$. Formally, this involves assigning an attribution score $\tau(\boldsymbol{x}, \boldsymbol{z}^{(n)}; \mathbb{S})$ to each training example, reflecting its relative importance in generating $\boldsymbol{x}$.

We follow [Zheng et al. (2024)](#) and adopt the linear datamodeling score (LDS; [Park et al., 2023a](#)) as our evaluation metric, which quantifies how well an attribution method aligns with the ground-truth influence of training data on model outputs. Specifically, for a training dataset $\mathbb{S} = \{\boldsymbol{z}^{(n)}\}_{n=1}^N$ of size $N$, LDS evaluates an attribution method $\tau$ by first sampling multiple random subsets $\{\mathbb{S}_m \subset \mathbb{S}\}_{m=1}^M$. Let $\theta^*(\mathbb{S}_m)$ denote the generative model trained on subset $\mathbb{S}_m$, and let $\mathcal{F}(\boldsymbol{x}; \theta^*(\mathbb{S}_m))$ denote the model output on a test input $\boldsymbol{x}$.[1] Then, for each subset $\mathbb{S}_m$, the attribution scores $\tau(\boldsymbol{x}, \boldsymbol{z}^{(n)}; \mathbb{S})$ assigned to training examples in $\mathbb{S}_m$ are summed to form an attribution-based prediction

---

[1] For diffusion models, we set $\mathcal{F} = \mathcal{L}_{\text{Simple}}$, representing the model output used in LDS evaluation.

$g_\tau(\boldsymbol{x}, \mathbb{S}_m; \mathbb{S})$:

$$g_\tau(\boldsymbol{x}, \mathbb{S}_m; \mathbb{S}) \triangleq \sum_{\boldsymbol{z}^{(n)} \in \mathbb{S}_m} \tau(\boldsymbol{x}, \boldsymbol{z}^{(n)}; \mathbb{S}). \tag{7}$$

Finally, the LDS score for attribution method $\tau$ on test input $\boldsymbol{x}$ is calculated as the Spearman rank correlation $\rho(\cdot, \cdot)$ between the ground-truth outputs from the $M$ retrained models (each trained on a different subset $\mathbb{S}_m$) and the corresponding attribution-based predictions:

$$\begin{aligned} \text{LDS}(\tau, \boldsymbol{x}) &\triangleq \\ \rho\Big(\{&\mathcal{F}(\boldsymbol{x}; \theta^*(\mathbb{S}_m))\}_{m \in [M]}, \{g_\tau(\boldsymbol{x}, \mathbb{S}_m; \mathbb{S})\}_{m \in [M]}\Big). \end{aligned} \tag{8}$$

## 3. Methodology

Our approach is motivated by the weighting term in the optimal score function of diffusion models, which naturally encodes the relative importance of each training example during generation. Leveraging this insight, we build upon the recent theoretical framework of [Kamb & Ganguli (2025)](#), which extends the analysis of optimal score functions to capture inductive biases that promote generalization in diffusion models. By bridging this framework with the problem of data attribution, we develop a *Nonparametric Data Attribution* (**NDA**) method that does not require access to model gradients or retraining.

### 3.1. Equivariant and Local Score Machines

While Eq. ([5](#)) defines the optimal score function, it relies solely on distance-based weighting in image space and does not generalize beyond the training set. [Kamb & Ganguli (2025)](#) derive an analytical form under inductive biases of *locality* and *equivariance*. This formulation preserves the structure of the empirical score but yields meaningful similarity by incorporating spatial structure and symmetry.

Let $\boldsymbol{x}_t \in \mathbb{R}^{C \times L \times L}$ denote a noisy image at diffusion time $t$, with $C$ channels and spatial resolution $L$. For a pixel location $\ell \in [L] \times [L]$, let $\boldsymbol{x}_{t,\ell} \in \mathbb{R}^C$ denote its pixel value. Define $\Omega_\ell$ as the $P \times P$ neighborhood centered at $\ell$, and $\boldsymbol{x}_{t,\Omega_\ell} \in \mathbb{R}^{C \times P \times P}$ as the corresponding local patch. Let $\mathbb{P}_\Omega(\mathbb{S})$ be the set of all such patches extracted from the training dataset $\mathbb{S}$. Each patch $\boldsymbol{u} \in \mathbb{P}_\Omega(\mathbb{S})$ is thus a local crop of some training image $\boldsymbol{z} \in \mathbb{S}$, and we denote its center pixel as $\boldsymbol{u}_0$. Under the locality and equivariance assumptions, [Kamb & Ganguli (2025)](#) show that the optimal MMSE estimator of the score function at pixel location $\ell$, denoted by $\boldsymbol{s}(\boldsymbol{x}_t, t, \ell) \in \mathbb{R}^C$, takes the form:

$$\boldsymbol{s}(\boldsymbol{x}_t, t, \ell) = \sum_{\boldsymbol{u} \in \mathbb{P}_\Omega(\mathbb{S})} \frac{\sqrt{\bar{\alpha}_t} \boldsymbol{u}_0 - \boldsymbol{x}_{t,\ell}}{1 - \bar{\alpha}_t} W_t(\boldsymbol{u} | \boldsymbol{x}_{t,\Omega_\ell}), \tag{9}$$

where the weighting term $W_t(\boldsymbol{u}|\boldsymbol{x}_{t,\Omega_\ell})$ is defined as:

$$W_t(\boldsymbol{u}|\boldsymbol{x}_{t,\Omega_\ell}) = \frac{\mathcal{N}\left(\boldsymbol{x}_{t,\Omega_\ell}|\sqrt{\bar{\alpha}_t}\boldsymbol{u}, (1-\bar{\alpha}_t)\,\mathbf{I}\right)}{\sum_{\boldsymbol{v}\in\mathbb{P}_\Omega(\mathbb{S})} \mathcal{N}\left(\boldsymbol{x}_{t,\Omega_\ell}|\sqrt{\bar{\alpha}_t}\boldsymbol{v}, (1-\bar{\alpha}_t)\,\mathbf{I}\right)}. \tag{10}$$

This formulation generalizes Eq. (5) by measuring similarity at the patch level rather than over full images, allowing it to exploit fine-grained local structure.

## 3.2. Patch-Based Data Attribution Scores

The weighting term in Eq. (10) can be expressed as a soft-max over quadratic distances $\frac{\|\boldsymbol{x}_{t,\Omega_\ell}-\sqrt{\bar{\alpha}_t}\boldsymbol{u}\|^2}{2(1-\bar{\alpha}_t)}$, which naturally reflects the contribution of each training patch $\boldsymbol{u}$ to the generation of pixel $\boldsymbol{x}_{t,\ell}$. We reinterpret this term as a *patch-wise influence* score that quantifies the influence of a training patch $\boldsymbol{u}$ on the local region of a generated image. By aggregating these local scores across spatial locations, we obtain a nonparametric, spatially interpretable attribution measure.

**Patch-wise influence.** For a noisy patch $\boldsymbol{x}_{t,\Omega_\ell}$ centered at $\ell$ in a generated image $\boldsymbol{x}$ at timestep $t$, we adopt the local weighting from Eq. (10) and define the patch-wise influence score as:

$$\tau(\boldsymbol{x}_{t,\Omega_\ell},\boldsymbol{u};\mathbb{P}_\Omega(\mathbb{S}))\triangleq\frac{\exp\left(-\frac{\left\|\boldsymbol{x}_{t,\Omega_\ell}-\sqrt{\bar{\alpha}_t}\boldsymbol{u}\right\|^2}{2(1-\bar{\alpha}_t)}\right)}{\sum_{\boldsymbol{v}\in\mathbb{P}_\Omega(\mathbb{S})}\exp\left(-\frac{\left\|\boldsymbol{x}_{t,\Omega_\ell}-\sqrt{\bar{\alpha}_t}\boldsymbol{v}\right\|^2}{2(1-\bar{\alpha}_t)}\right)}, \tag{11}$$

which is a normalized similarity score over all patches in $\mathbb{P}_\Omega(\mathbb{S})$ extracted from the training set.

**Image-level attribution.** Given a generated image $\boldsymbol{x}$, we apply the forward diffusion process to obtain noisy samples $\boldsymbol{x}_t$. For each pixel location $\ell$, we extract the $P\times P$ patch $\boldsymbol{x}_{t,\Omega_\ell}$ from $\boldsymbol{x}_t$, using zero-padding near boundaries to handle incomplete patches. The patch-wise influence scores in Eq. (11) are then computed for all training patches $\boldsymbol{u}\in\mathbb{P}_\Omega(\mathbb{S})$.

To aggregate them into an image-level attribution score $\tau(\boldsymbol{x},\boldsymbol{z}^{(n)};\mathbb{S})$, we proceed as follows: (1) For each patch $\boldsymbol{x}_{t,\Omega_\ell}$ of $\boldsymbol{x}_t$, we select the $k$ most influential patches from each training image $\boldsymbol{z}^{(n)}$, denoted by $\mathbb{P}_\Omega^k(\boldsymbol{x}_{t,\Omega_\ell},\boldsymbol{z}^{(n)})$. [2] (2) We sum the influence scores of these top-$k$ patches to estimate the contribution of the training image $\boldsymbol{z}^{(n)}$ to generating the local region around $\ell$. (3) Finally, we aggregate across all spatial locations $\ell$ and average over a set of

---

[2]Formally, the selected set is defined precisely as the top-$k$ subset of image patches under the patch-wise influence score, that is to say, $\mathbb{P}_\Omega^k(\boldsymbol{x}_{t,\Omega_\ell},\boldsymbol{z}^{(n)})\subset\mathbb{P}_\Omega(\{\boldsymbol{z}^{(n)}\})$ together with the cardinality constraint $|\mathbb{P}_\Omega^k(\boldsymbol{x}_{t,\Omega_\ell},\boldsymbol{z}^{(n)})|=k$, and for all of the selected patches $\boldsymbol{u}'\in\mathbb{P}_\Omega^k(\boldsymbol{x}_{t,\Omega_\ell},\boldsymbol{z}^{(n)})$ and all of the remaining unselected patches $\boldsymbol{u}''\in\mathbb{P}_\Omega(\{\boldsymbol{z}^{(n)}\})\setminus\mathbb{P}_\Omega^k(\boldsymbol{x}_{t,\Omega_\ell},\boldsymbol{z}^{(n)})$, we have the score ordering $\tau(\boldsymbol{x}_{t,\Omega_\ell},\boldsymbol{u}';\mathbb{P}_\Omega(\mathbb{S}))\geq\tau(\boldsymbol{x}_{t,\Omega_\ell},\boldsymbol{u}'';\mathbb{P}_\Omega(\mathbb{S}))$.

timesteps $\mathcal{T}$:

$$\tau(\boldsymbol{x},\boldsymbol{z}^{(n)};\mathbb{S})\triangleq\frac{1}{|\mathcal{T}|}\sum_{t\in\mathcal{T}}\sum_{\ell}\sum_{\substack{\boldsymbol{u}\in\\\mathbb{P}_\Omega^k(\boldsymbol{x}_{t,\Omega_\ell},\boldsymbol{z}^{(n)})}}\tau(\boldsymbol{x}_{t,\Omega_\ell},\boldsymbol{u};\mathbb{P}_\Omega(\mathbb{S})). \tag{12}$$

This attribution score is both *spatially interpretable*, as it aggregates patch-wise influence with local meaning, and *nonparametric*, in the sense that it is computed from the training data without using target-model parameters or gradients.

## 3.3. Attribution with Multiscale Patch-Wise Influence

The attribution method described above relies on quadratic Euclidean distances between fixed-size patches in the original image space. However, images may have varying resolutions, and a single patch size may fail to capture different levels of information: from fine-grained textures to higher-level structures (Adelson et al., 1984; Lin et al., 2017). Moreover, during denoising, early high-noise stages capture coarse structures, while later low-noise stages refine local details (Jing et al., 2022; Park et al., 2023b).

To account for these effects, we introduce a *multiscale* extension that computes patch-wise influence across multiple resolutions. Specifically, we downsample both generated and training patches and evaluate distances in the lower-resolution space. Let $D(\cdot)$ denote a downsampling operator, and define $\widehat{\boldsymbol{x}}_{t,\Omega_\ell}\triangleq D(\boldsymbol{x}_{t,\Omega_\ell})$, $\widehat{\boldsymbol{u}}\triangleq D(\boldsymbol{u})$, and $\widehat{\boldsymbol{v}}\triangleq D(\boldsymbol{v})$. For the non-overlapping $2\times 2$ average pooling used in our experiments, $DD^\top=c_D I$ with $c_D=1/4$. We therefore define the low-resolution patch-wise influence score as:

$$\widehat{\tau}(\boldsymbol{x}_{t,\Omega_\ell},\boldsymbol{u};\mathbb{P}_\Omega(\mathbb{S}))\triangleq\frac{\exp\left(-\frac{\left\|\widehat{\boldsymbol{x}}_{t,\Omega_\ell}-\sqrt{\bar{\alpha}_t}\widehat{\boldsymbol{u}}\right\|^2}{2c_D(1-\bar{\alpha}_t)}\right)}{\sum_{\boldsymbol{v}\in\mathbb{P}_\Omega(\mathbb{S})}\exp\left(-\frac{\left\|\widehat{\boldsymbol{x}}_{t,\Omega_\ell}-\sqrt{\bar{\alpha}_t}\widehat{\boldsymbol{v}}\right\|^2}{2c_D(1-\bar{\alpha}_t)}\right)}. \tag{13}$$

We then combine the original and low-resolution influence measures into a multiscale score:

$$\begin{aligned}&\tau^{\mathrm{ms}}(\boldsymbol{x}_{t,\Omega_\ell},\boldsymbol{u};\mathbb{P}_\Omega(\mathbb{S}))\triangleq\\&\gamma_t\tau(\boldsymbol{x}_{t,\Omega_\ell},\boldsymbol{u};\mathbb{P}_\Omega(\mathbb{S}))+(1-\gamma_t)\widehat{\tau}(\boldsymbol{x}_{t,\Omega_\ell},\boldsymbol{u};\mathbb{P}_\Omega(\mathbb{S})),\end{aligned} \tag{14}$$

where $\gamma_t\in[0,1]$ is a timestep-dependent weighting factor that balances the contribution of the original and low-resolution influence. In Eq. (15), $\mathbb{P}_\Omega^k(\boldsymbol{x}_{t,\Omega_\ell},\boldsymbol{z}^{(n)})$ denotes the top-$k$ patches selected according to the multiscale score $\tau^{\mathrm{ms}}$. Finally, we extend this to multiscale image-level attribution by aggregating over timesteps $\mathcal{T}$ and spatial locations $\ell$:

$$\begin{aligned}\tau^{\mathrm{ms}}(\boldsymbol{x},\boldsymbol{z}^{(n)};\mathbb{S})\triangleq\frac{1}{|\mathcal{T}|}\sum_{t\in\mathcal{T}}\sum_{\ell}\sum_{\boldsymbol{u}\in\mathcal{P}_{t,\ell,n}^k}\\\tau^{\mathrm{ms}}(\boldsymbol{x}_{t,\Omega_\ell},\boldsymbol{u};\mathbb{P}_\Omega(\mathbb{S}))\,.\end{aligned} \tag{15}$$

### 3.4. Convolution-Based Acceleration

Directly evaluating Eqs. (11) and (13) on large-scale datasets is computationally challenging. In a naive implementation, each test patch $x_{t,\Omega_\ell}$ is broadcast against all unfolded training patches $u \in \mathbb{P}_\Omega(\mathbb{S})$, leading to peak memory consumption of $\mathcal{O}(NL^2CP^2)$, which is $P^2$ times larger than the dataset itself.

To address this, we propose a memory-efficient implementation that avoids explicit patch unfolding by leveraging convolutional operators. Specifically, to compute $\|x_{t,\Omega_\ell} - \sqrt{\bar{\alpha}_t}u\|^2$ for all $L \times L$ patches $u$ from a training image $z^{(n)}$, we treat the patch $x_{t,\Omega_\ell} \in \mathbb{R}^{C \times P \times P}$ as a convolutional kernel. Applying this kernel to the training image yields the inner-products $\langle x_{t,\Omega_\ell}, u \rangle$ over all spatial locations in a single convolution pass. The quadratic distance is then obtained as $\|x_{t,\Omega_\ell}\|^2 - 2\sqrt{\bar{\alpha}_t}\langle x_{t,\Omega_\ell}, u \rangle + \bar{\alpha}_t\|u\|^2$.

This approach leverages GPU-optimized convolutions to reduce memory usage. We further parallelize across the $L^2$ test patches by batching $B$ patches into a convolutional kernel with $B$ output channels. This yields peak memory $\mathcal{O}(BNL^2)$, which does not explicitly scale with patch size $P$, thereby avoiding the prohibitive $P^2$ factor of naive unfolding and enabling scalable attribution on large datasets.

### 3.5. Discussion

Data attribution is often regarded as a lens for understanding model behavior, quantifying the causal influence of individual training examples on model predictions through the learning process. From this perspective, it might seem paradoxical to speak of attribution without access to target-model parameters or gradients: if no such access is available, what exactly is being attributed?

Our work invites a broader interpretation. In a narrow sense, our nonparametric method can be understood as a principled estimate of how training data might have influenced a generated sample, assuming it was produced by some model trained on the same data. Since our method does not access model parameters, it produces the same attribution for all possible models trained on the dataset. At first glance, this may appear counterintuitive, because different models can generalize in different ways and might be expected to yield different attribution patterns. However, our empirical results show that this attribution signal remains useful under the tested changes in model architecture and training objective (see Appendix D.1). This suggests that, in these settings, patch-level relationships between training data and generated outputs provide useful information for attribution, even without access to target-model parameters or gradients.

Seen from an even broader perspective, this idea resonates with how humans attribute provenance. People are often able to determine which training images most likely inspired a generated image even without any knowledge of the mechanism that produced it. This form of attribution is grounded in *perceived similarity* rather than *parametric causality*. In this light, nonparametric data attribution can be viewed as an attempt to formalize this intuitive notion of influence: some training examples are more strongly associated with a given generation than others.

## 4. Experiments

We evaluate our method, *Nonparametric Data Attribution* (**NDA**), against both nonparametric and gradient-based attribution approaches using two complementary protocols: the linear datamodeling score (LDS), which quantifies alignment with ground-truth influence, and counterfactual evaluation, which assesses the effect of removing influential training data on generated outputs. Our results and ablations show that NDA achieves strong attribution performance without accessing model parameters or gradients, while qualitative visualizations highlight its spatial interpretability and visual consistency.

### 4.1. Experimental Setup

**Datasets.** We use CIFAR-10 (Krizhevsky, 2009) and CelebA (Liu et al., 2015). For efficient ablation studies, we follow Zheng et al. (2024) and construct a CIFAR-2 subset consisting of 5,000 images from CIFAR-10. We additionally use ArtBench-2 (Liao et al., 2022) at 256×256 resolution. Additional dataset details are provided in Appendix B.1.

**Target models.** For each dataset, we train a diffusion model to serve as the target model for data attribution. On CIFAR-2 and CIFAR-10, we adopt the original DDPM implementation (Ho et al., 2020) to train an unconditional diffusion model with a U-Net backbone containing 35.7M parameters. For CelebA, we use the same implementation but with a modified architecture of 118.8M parameters to accommodate the 64×64 resolution. For ArtBench-2 at 256×256, we follow the target-model setup of Zheng et al. (2024). The number of diffusion steps is fixed to $T$=1,000. Further training details are provided in Appendix B.2. Note that gradient-based attribution methods directly use the target model parameters, whereas our NDA does not access model parameters or architectures.

**LDS evaluation.** Following Zheng et al. (2024), we sample $M$=64 random subsets of the training set $\mathbb{S}$, each containing 50% of the samples. For each subset $\mathbb{S}_m$, we train three models with different random seeds and average their $\mathcal{L}_{\text{Simple}}$ losses as the model output $\mathcal{F}(x; \theta^*(\mathbb{S}_m))$ for a test input $x$. Additional implementation details are provided in Appendix B.3. To ensure a fair comparison, we use the same 1,000-image held-out validation set and 1,000-image

*Table 1.* LDS (%) of different attribution methods on CIFAR-2, CIFAR-10, and CelebA.

| Method | CIFAR-2 | | CIFAR-10 | | CelebA | |
|---|---|---|---|---|---|---|
| | Validation | Generation | Validation | Generation | Validation | Generation |
| *Without Model Gradients* | | | | | | |
| Raw pixel (dot prod.) | 7.77±0.57 | 4.89±0.58 | 2.50±0.42 | 2.25±0.39 | 5.58±0.73 | 4.94±1.58 |
| Raw pixel (cosine) | 7.87±0.57 | 5.44±0.57 | 2.71±0.41 | 2.61±0.38 | 6.16±0.75 | 4.38±1.63 |
| CLIP similarity (dot prod.) | 6.51±1.06 | 3.00±0.95 | 2.39±0.41 | 1.11±0.47 | 8.87±1.14 | 2.51±1.13 |
| CLIP similarity (cosine) | 8.54±1.01 | 4.01±0.85 | 3.39±0.38 | 1.69±0.49 | 10.92±0.87 | 3.03±1.13 |
| **NDA (Ours)** | **24.88±0.42** | **15.91±0.49** | **11.81±0.30** | **7.41±0.45** | **16.89±0.59** | **13.92±0.68** |
| *Using Model Gradients* | | | | | | |
| Gradient (dot prod.) | 5.14±0.60 | 2.80±0.55 | 0.79±0.43 | 0.74±0.45 | 3.82±0.50 | 3.83±1.06 |
| Gradient (cosine) | 5.08±0.59 | 2.78±0.54 | 0.66±0.43 | 0.58±0.42 | 3.65±0.52 | 3.86±0.96 |
| TracInCP | 6.26±0.84 | 3.76±0.61 | 0.98±0.44 | 0.96±0.40 | 5.14±0.75 | 5.18±1.05 |
| GAS | 5.78±0.82 | 3.34±0.56 | 0.89±0.48 | 0.90±0.41 | 5.44±0.68 | 4.69±0.97 |
| TRAK | 11.42±0.49 | 5.78±0.48 | 2.93±0.46 | 2.20±0.38 | 11.28±0.47 | 7.02±0.89 |
| D-TRAK | 26.79±0.33 | 18.82±0.43 | 14.69±0.46 | 11.05±0.43 | 22.83±0.51 | 16.84±0.54 |

*Figure 2.* Counterfactual evaluation of $\ell_2$ distance (**Left**) and CLIP similarity (**Right**) between original and regenerated images on CIFAR-2 and CelebA after removing the most influential training samples identified by different attribution methods and retraining the model.

generation set as Zheng et al. (2024).

**NDA setup.** For patch-wise influence, we use patch sizes $P \in [3, 21]$ for different timesteps $t \in \mathcal{T}$ and resolutions, as determined by ablation studies in Sec. 4.5. To compute low-resolution patch-wise influence, we apply a non-overlapping average-pooling downsampling operator $D(\cdot)$ with a window size of 2, reducing patch resolution by half. In all experiments, we select the top-$k$ most influential patches per training image with $k=100$ to obtain image-level attribution scores. Due to low signal-to-noise ratios at large timesteps, we restrict the set of timesteps to $\mathcal{T}=\{100, 200, 300, 400, 500\}$.

### 4.2. Main Results: LDS Evaluation

We compare NDA against a range of attribution baselines; detailed descriptions are provided in Appendix B.4. Our primary comparison focuses on representative approaches that do not require access to target-model parameters or gradients, with gradient-based methods included as references. Table 1 reports the results on CIFAR-2, CIFAR-10, and CelebA.

Compared with CLIP cosine similarity, a strong nonparametric baseline, NDA achieves consistent and substantial

gains across both validation and generation sets. On CIFAR-2, NDA improves over CLIP by +16.34 (validation) and +11.90 (generation); on CIFAR-10, by +8.42 and +5.72; and on CelebA, by +5.97 and +10.89, respectively. When compared with gradient-based methods tailored for diffusion models, NDA remains below D-TRAK but reduces the gap relative to CLIP similarity while requiring no access to target-model parameters or gradients. For example, on CelebA, NDA trails D-TRAK by 5.94/2.92 (validation/ generation), whereas CLIP cosine similarity trails D-TRAK by 11.91/13.81. Overall, NDA markedly outperforms existing nonparametric baselines on LDS. Among the evaluated methods that do not use target-model parameters or gradients, NDA achieves the best LDS performance.

We further evaluate NDA at 256×256 resolution on ArtBench-2 (Liao et al., 2022), a two-style subset of Art-Bench in a style-conditioned setting. For NDA, we use stride-2 spatial patch subsampling; Appendix F reports a supporting stride study on CelebA. Following the style-conditioned evaluation protocol, we restrict the candidate pool for attribution to ArtBench-2 training images with the same style label as the evaluated image. For generated samples, we use the conditioning label as the style label. As shown in Table 2, NDA achieves 14.35/12.98 LDS on

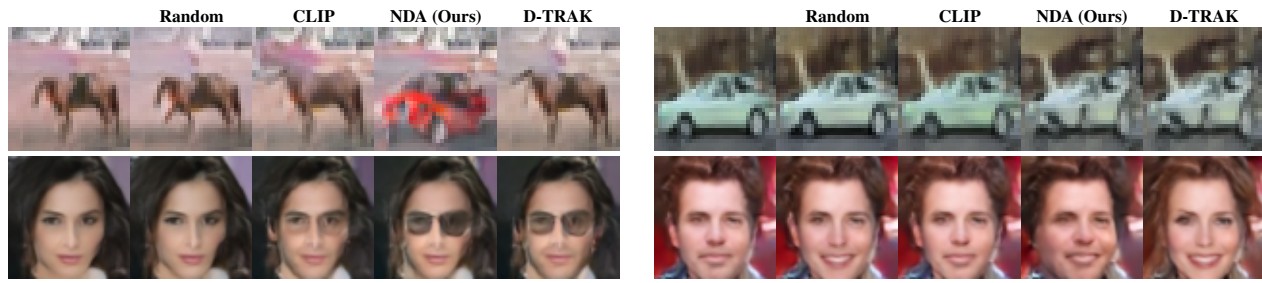

*Figure 3.* Counterfactual visualization on CIFAR-2 (**Top**) and CelebA (**Bottom**). Images are compared to those generated by retrained models using the same seed. See Appendix G.1 for more cases.

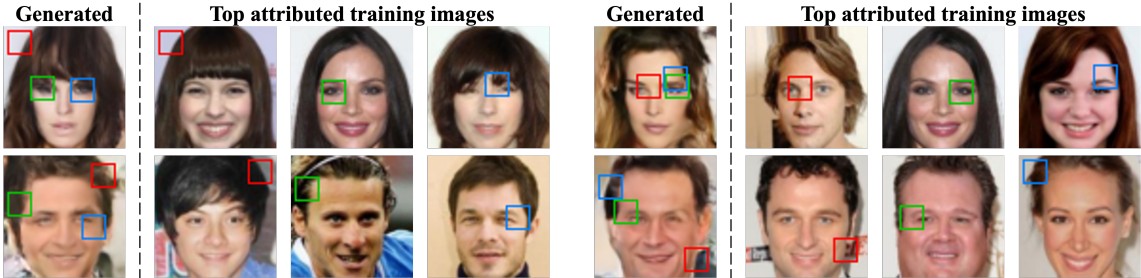

*Figure 4.* Spatial interpretability of NDA. The patches with the highest patch-wise influence scores (w.r.t. patches in the generated image) are highlighted in the top attributed training images.

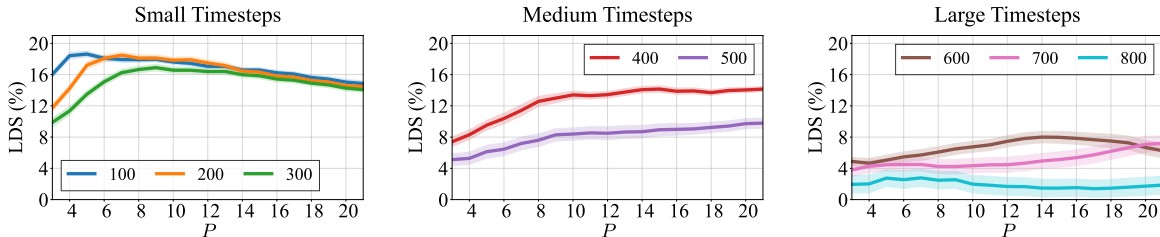

*Figure 5.* **Patch size vs. timestep.** LDS (%) on CIFAR-2 validation across patch sizes and timesteps at the original resolution. **Left**: $t \leq 300$. **Middle**: $t = 400, 500$. **Right**: $t \geq 600$, where noise dominates.

validation/generation, outperforming Raw pixel cosine similarity (2.58/2.71), CLIP cosine similarity (8.62/8.66), and TRAK (12.26/7.78). D-TRAK obtains 27.61/24.16 under the same protocol, leaving a clear gap in this 256×256 setting. Among the evaluated methods that do not use target-model parameters or gradients, NDA gives the best LDS results at 256×256.

### 4.3. Counterfactual Evaluation

To evaluate the faithfulness and practical effectiveness of NDA, we conduct a counterfactual influence experiment on CIFAR-2 and CelebA. For each generated test image, we first identify the top-1,000 most positively influential training samples according to each attribution method, remove these samples from the training set, and retrain the diffusion model from scratch. The test image is then regenerated using the retrained models under the same random seed, and

the impact of removal is quantified using pixel-wise $\ell_2$ distance and CLIP cosine similarity. We repeat this procedure for 60 randomly generated test images and summarize the results with boxplots.

We compare NDA against three baselines: Random removal, CLIP similarity (cosine), and D-TRAK. As shown in Fig. 2, NDA achieves median $\ell_2$ distances of $8.43$ and $14.38$ on CIFAR-2 and CelebA, respectively, outperforming CLIP $(6.68, 10.30)$ and D-TRAK $(7.68, 13.18)$. For CLIP similarity, NDA attains $0.92$ and $0.94$, approaching D-TRAK $(0.91, 0.93)$. These results indicate that NDA effectively identifies and removes training samples with influence over the generated output, providing a practical nonparametric alternative when target-model gradients are unavailable. The qualitative visualizations in Fig. 3 further show that the generated images exhibit noticeable changes after removing the training samples deemed most influential by NDA.

*Table 2.* LDS (%) of different attribution methods on ArtBench-2.

| Method | Validation | Generation |
|---|---|---|
| *Without Model Gradients* | | |
| Raw pixel (dot prod.) | 2.44±0.56 | 2.60±0.84 |
| Raw pixel (cosine) | 2.58±0.56 | 2.71±0.86 |
| CLIP similarity (dot prod.) | 7.18±0.70 | 5.33±1.45 |
| CLIP similarity (cosine) | 8.62±0.70 | 8.66±1.31 |
| **NDA (Ours)** | **14.35±0.50** | **12.98±0.55** |
| *Using Model Gradients* | | |
| Gradient (dot prod.) | 7.68±0.43 | 4.07±1.07 |
| Gradient (cosine) | 7.72±0.42 | 4.50±0.97 |
| TracInCP | 9.69±0.49 | 6.36±0.93 |
| GAS | 9.65±0.46 | 6.74±0.82 |
| TRAK | 12.26±0.42 | 7.78±0.59 |
| D-TRAK | 27.61±0.49 | 24.16±0.67 |

## 4.4. Spatial Interpretability

Since NDA computes attribution by aggregating patch-wise influence, it naturally offers an additional spatial level of interpretability. Intuitively, a training image is assigned a higher attribution score when its local patches align well with regions of the generated image. To visualize this correspondence, we highlight the most influential patches from each of the top-attributed training images in Fig. 4. This visualization shows which patch-level correspondences contribute to the image-level attribution score. In these examples, top-ranked training images contain local patches that are visually similar to those in the generated image, offering a natural explanation for their high attribution scores.

## 4.5. Ablation Studies

We conduct ablation studies on key hyperparameters of our method. All studies are performed on the validation sets, which are i.i.d. samples drawn from the same distribution as the training sets. The selected hyperparameters are then applied to the generation sets without further tuning to produce the results reported in Sec. 4.2. In all figures where applicable, solid lines show the mean, and shaded regions indicate ±1 standard deviation.

**Patch size selection.** We study the effect of patch size by evaluating LDS performance across $P \in [3, 21]$ at different timesteps. To isolate the impact of timestep, we fix $\mathcal{T}$ to contain a single timestep (e.g., $\mathcal{T}=\{100\}, \ldots, \{900\}$) and vary $P$. Fig. 5 shows LDS curves as a function of $P$ for each $\mathcal{T}$ on CIFAR-2 (see Appendix C for CelebA). We observe that the optimal patch size varies with timestep and generally increases as $t$ grows. At small forward timesteps ($t \leq 300$), small to moderate patches ($P=5, 7, 9$) yield the highest LDS scores, suggesting that local patterns dominate when noise levels are relatively low. For mid-range timesteps ($400 \leq t \leq 500$), larger patches perform better, due to their ability to aggregate more contextual information

under higher noise levels. In the high-noise regime ($t \geq 600$), LDS scores drop significantly across all patch sizes. Based on these results, we adopt a timestep-dependent patch size selection strategy: using smaller patches for small forward timesteps and progressively larger patches for larger forward timesteps, choosing the $P$ that maximizes LDS for each $t$.

**Multiscale influence.** We first find the optimal patch size for the low-resolution patch-wise influence in Eq. (13) using the same procedure as above and observe a similar trend, where small forward timesteps favor smaller patches. Interestingly, the optimal patch size differs between the original and low-resolution cases (see Appendix C for details). We then evaluate the proposed multiscale attribution in Eq. (15) by varying the weighting factor $\gamma \in \{0, 0.25, 0.5, 0.75, 1\}$ across timesteps. As shown in Fig. 6, combining multiple scales (e.g., $\gamma=0.75$) consistently improves LDS, particularly in the low-noise regime, suggesting that multiscale features provide complementary information that enhances attribution. Further improvements may be possible by incorporating more scales, which we leave for future work.

**Top-$k$ patch selection.** We examine how the number of top-$k$ patches affects attribution. Fig. 7 shows LDS as a function of $k$ on lower-noise timesteps, which contribute more significantly to the final attribution. We test $k$ ranging from 1 to 500 and find that an intermediate value of $k=100$ generally achieves optimal performance across timesteps and datasets and generalizes well to generation sets.

**Timesteps.** As lower-noise timesteps provide stronger attribution signal, we define $\mathcal{T}=\{100, 200, \ldots, \tau_{\max}\}$ and study the effect of varying the max aggregation timestep $\tau_{\max}$. At each $t \in \mathcal{T}$ we use the optimal patch sizes and weighting factors identified above. Fig. 8 shows that enlarging $\mathcal{T}$ improves LDS up to mid-range timesteps, after which gains saturate or degrade. On CIFAR-2, performance rises from $t=100$ to a peak near $\tau_{\max}=400$ before dropping by $\sim 0.5$ at $\tau_{\max} \geq 600$; on CelebA, performance saturates for $\tau_{\max}$ between 400 and 500 with negligible gains beyond. These results suggest that late (high-noise) timesteps contribute little useful signal and may instead inject noise. We therefore fix $\tau_{\max}=500$ for all subsequent experiments, which achieves near-optimal performance while avoiding unnecessary computation and variability.

## 5. Conclusion

We present a nonparametric method for data attribution in diffusion models that measures the influence of training examples via patch-level similarity. Our approach leverages the analytical form of the optimal score function, extends to multiscale representations, and remains efficient through convolution-based operations. It provides spatially interpretable attributions and captures local correspondences

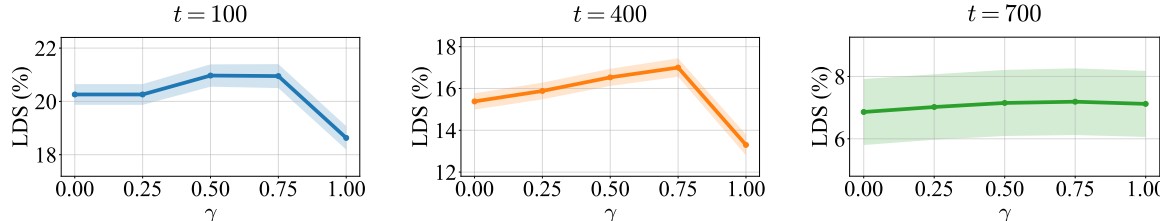

Figure 6. LDS (%) on CIFAR-2 across different values of the weighting factor $\gamma$ at varying timesteps.

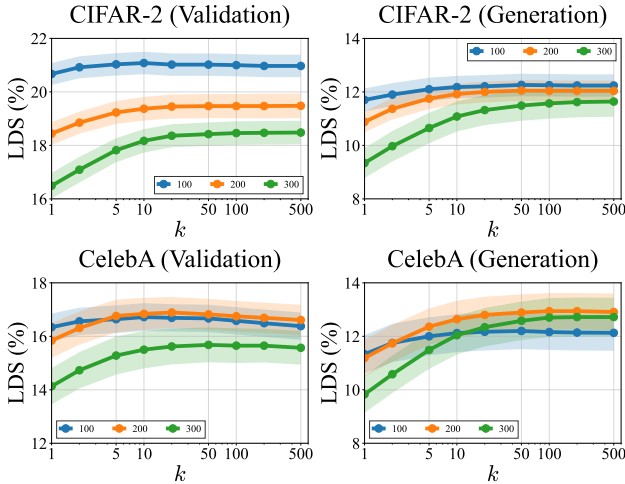

Figure 7. LDS (%) as a function of the number $k$ of top influential patches selected for aggregation on CIFAR-2 (**Top**) and CelebA (**Bottom**). Setting $k{=}100$ generally provides strong performance.

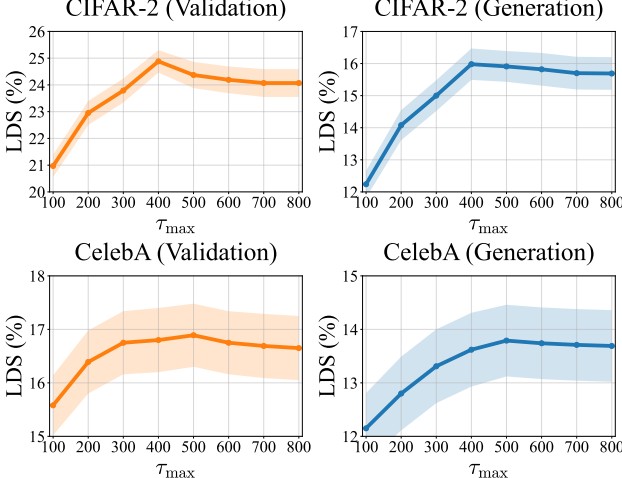

Figure 8. LDS (%) as a function of the max aggregation timestep $\tau_{\max}$ on CIFAR-2 (**Top**) and CelebA (**Bottom**). Choosing $\tau_{\max} = 500$ generally yields good results.

between training data and generated outputs. Experiments on CIFAR-2, CIFAR-10, CelebA, and ArtBench-2 show that our method outperforms the existing nonparametric baselines and achieves the best LDS performance among the evaluated methods that do not require target-model parameters or gradients, suggesting that such data-driven attribution can be practical for training-data influence analysis in diffusion models.

## Impact Statement

This work aims to improve data transparency for generative models. Our method is motivated by attribution settings where target-model parameters or gradients may be unavailable, including copyright auditing, data valuation, and analyses of training-data use. Like other attribution methods, NDA assigns scores that estimate the influence of training examples rather than providing direct proof of data use. These scores should therefore be interpreted with care and should not be used as standalone evidence in legal, copyright, or data valuation decisions.

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

# A. Related Work

## A.1. Data Attribution

Training data plays a critical role in shaping the behavior of machine learning models. Data attribution aims to measure the contribution of individual training samples to model predictions. Existing methods can be broadly categorized into two classes: retraining-based and retraining-free approaches.

Retraining-based methods, such as empirical influence functions (Feldman & Zhang, 2020), Shapley value estimators (Jia et al., 2021), and datamodels (Ilyas et al., 2022), provide high-fidelity attributions by measuring the effect of removing or modifying each training example. However, these methods typically require retraining the model tens of thousands of times on different data subsets to achieve reliable results, making them computationally infeasible for large-scale settings. Retraining-free methods aim to approximate influence scores without additional retraining, offering a more scalable alternative. These approaches fall into two main categories. The first comprises gradient-based methods without kernelization, which rely solely on first-order gradient signals and avoid computing second-order derivatives. Representative approaches include Gradient (Charpiat et al., 2019), TracInCP (Pruthi et al., 2020) and GAS (Hammoudeh & Lowd, 2022). The second category includes gradient-based methods with kernelization, which incorporate curvature information by constructing kernels, typically using the inverse of the Hessian matrix. However, Hessian inversion is often numerically unstable. To mitigate this, recent work approximates the Hessian with the Fisher information matrix (FIM). For instance, TRAK (Park et al., 2023a) introduces a kernel-based approximation that efficiently estimates influence scores via random projections, which is both accurate and computationally tractable for large-scale models. D-TRAK (Zheng et al., 2024) further improves attribution performance by modifying the output function and training loss in TRAK, resulting in more effective LDS scores. Based on empirical analysis, D-TRAK recommends a specific configuration that achieves the best overall performance.

## A.2. Nonparametric Data Attribution Methods

While most data attribution methods rely on parametric models and require gradient computations or model retraining, an alternative line of research explores nonparametric approaches that operate directly on training data without fitting parametric functions. These are typically similarity-based approaches, which estimate the influence of a training sample based on its similarity to the target using a predefined metric. For example, one can leverage pretrained vision-language embeddings such as CLIP (Radford et al., 2021) features to measure high-level semantic similarity of images (Zheng et al., 2024; Mlodozeniec et al., 2025), or compare raw pixel values directly. However, these methods overlook the internal dynamics of the generative model. As a result, effective data attribution for diffusion models in restricted-access settings where target-model parameters or gradients are unavailable remains an open and challenging problem.

# B. Implementation Details

## B.1. Datasets

**CIFAR-10** ($32 \times 32$)**.** The CIFAR-10 dataset contains 50,000 training samples. For LDS evaluation, we randomly sample 1,000 validation images from the CIFAR-10 test set. To reduce computational overhead, we also construct a subset called CIFAR-2, consisting of 5,000 training images randomly selected from the "automobile" and "horse" classes in CIFAR-10's training set, along with 1,000 validation images from the corresponding classes in the test set.

**CelebA** ($64 \times 64$)**.** We constructed our dataset by selecting 5,000 training samples and 1,000 validation samples from the original training and test splits of CelebA (Liu et al., 2015). Following the preprocessing steps outlined by (Song et al., 2021), we first center-crop the images to $140 \times 140$ pixels, and then resize them to $64 \times 64$ pixels.

**ArtBench-2** ($256 \times 256$)**.** ArtBench (Liao et al., 2022) contains 60,000 artwork images from 10 artistic styles, with 5,000 training images and 1,000 test images per style. Following Zheng et al. (2024), we use the two-style ArtBench-2 subset drawn from *post-impressionism* and *ukiyo-e*. Our evaluation uses 5,000 training images, 1,000 held-out validation images, and 1,000 generated images from the corresponding target model, all at $256 \times 256$ resolution.

## B.2. Models

**CIFAR.** We follow the original implementation of the unconditional DDPM (Ho et al., 2020), which uses a U-Net backbone comprising approximately 35.7M parameters. The model is trained for 200 epochs with a batch size of 128, using a cosine annealing learning rate schedule. Additional model configurations include a dropout rate of 0.1 and the use of the AdamW

optimizer (Loshchilov & Hutter, 2019) with a weight decay of $10^{-6}$. To enhance robustness, random horizontal flips are applied as a data augmentation strategy.

**CelebA.** We adopted an unconditional DDPM framework similar to that used for CIFAR-10, but modified the architecture to handle $64 \times 64$ inputs. To better capture the higher complexity of CelebA, we scaled up the U-Net model to $118.8M$ parameters. All training settings, including the noise variance schedule, optimizer configurations, and overall training procedure, were kept consistent with those used in the CIFAR-10 setup.

**ArtBench-2.** For ArtBench-2, following the $256 \times 256$ target-model setup of Zheng et al. (2024), we use a LoRA-finetuned Stable Diffusion model adapted to $256 \times 256$ resolution. Class conditioning is specified through simple style prompts, such as "a {style} painting". All other training and sampling settings are kept consistent with Zheng et al. (2024).

### B.3. LDS Evaluation Setup

For the LDS evaluation, we sample $M = 64$ random subsets $\mathbb{S}_m$ from the training set, with each subset comprising $50\%$ of the data (i.e., $\alpha = 0.5$). For each subset, three models are trained using different random seeds to improve robustness. We then compute the linear datamodeling score for each sample of interest as the Spearman rank correlation between the model output and the attribution score as described in Eq. (8). In particular, to compute the simple loss $\mathcal{L}_{\text{Simple}}(\boldsymbol{x}, \theta)$ as defined in Eq. (3) for any sample of interest, we evaluate it over 1000 timesteps uniformly spaced in the interval $[1, T]$. At each timestep, we further approximate the expectation $\mathbb{E}_{\epsilon}$ by sampling three standard Gaussian noise vectors $\boldsymbol{\epsilon} \sim \mathcal{N}(0, \mathbf{I})$. Finally, we average the LDS scores across the validation and generation samples to obtain the overall performance.

### B.4. Baselines

We compare our proposed method against two major categories of attribution methods: (1) nonparametric similarity-based methods, which do not rely on model parameters and instead operate directly on image features or representations, and (2) post-hoc retraining-free methods, which leverage gradient-based representations of a trained model to estimate attribution scores without requiring model retraining. Within the first category, we include two similarity-based methods: Raw Pixel and CLIP (Radford et al., 2021). Within the second category, we further divide the methods into *gradient-based methods without kernels* and *gradient-based methods with kernels*. For the gradient-based methods without kernels, we include techniques such as Gradient (Charpiat et al., 2019), TracInCP (Pruthi et al., 2020), and GAS (Hammoudeh & Lowd, 2022). As for the gradient-based methods with kernels, we compare against representative methods including TRAK (Park et al., 2023a) and D-TRAK (Zheng et al., 2024).

**Raw Pixel.** This is a naive similarity-based attribution method that directly uses the raw image as the feature representation. The attribution score is then computed by measuring the similarity—such as the dot product or cosine similarity—between the query sample and each training sample in the dataset.

**CLIP Similarity.** We adopt CLIP (Radford et al., 2021) as a baseline attribution method. Each target sample and training sample is embedded using CLIP. We report both the dot product of the raw embeddings and the cosine similarity of normalized embeddings as attribution scores. This provides a simple way to estimate the influence of training data, independent of the diffusion model or the LDS measurement function.

**Gradient.** This method is a gradient-based influence estimator from (Charpiat et al., 2019), where the attribution score for each training sample is computed using either the dot product or cosine similarity between its gradient (with respect to the model parameters) and that of a given test sample.

$$\tau(\boldsymbol{x}, \boldsymbol{z}^{(n)}; \mathbb{S}) = \left(\mathcal{P}^{\top} \nabla_{\theta} \mathcal{L}_{\text{Simple}}(\boldsymbol{x}, \theta^*)\right)^{\top} \cdot \mathcal{P}^{\top} \nabla_{\theta} \mathcal{L}_{\text{Simple}}(\boldsymbol{z}^{(n)}, \theta^*)$$

$$\tau(\boldsymbol{x}, \boldsymbol{z}^{(n)}; \mathbb{S}) = \frac{\left(\mathcal{P}^{\top} \nabla_{\theta} \mathcal{L}_{\text{Simple}}(\boldsymbol{x}, \theta^*)\right)^{\top} \cdot \mathcal{P}^{\top} \nabla_{\theta} \mathcal{L}_{\text{Simple}}(\boldsymbol{z}^{(n)}, \theta^*)}{\left\|\mathcal{P}^{\top} \nabla_{\theta} \mathcal{L}_{\text{Simple}}(\boldsymbol{x}, \theta^*)\right\| \left\|\mathcal{P}^{\top} \nabla_{\theta} \mathcal{L}_{\text{Simple}}(\boldsymbol{z}^{(n)}, \theta^*)\right\|}.$$

where $\mathcal{P}$ is the Gaussian random projection matrix that projects the gradient into a low-dimensional subspace.

**TracInCP.** We adopt the TracInCP estimator introduced by (Pruthi et al., 2020) formulated as $\tau(\boldsymbol{x}, \boldsymbol{z}^{(n)}; \mathbb{S}) = \frac{1}{C} \sum_{c=1}^{C} \left(\mathcal{P}_c^{\top} \nabla_{\theta} \mathcal{L}_{\text{Simple}}(\boldsymbol{x}, \theta^c)\right)^{\top} \cdot \left(\mathcal{P}_c^{\top} \nabla_{\theta} \mathcal{L}_{\text{Simple}}(\boldsymbol{z}^{(n)}, \theta^c)\right)$ where $C$ denotes the number of model checkpoints uniformly sampled from the training trajectory, and $\theta^c$ represents the parameters at the $c$-th checkpoint. We use four checkpoints for each experiment. For example, on CIFAR-2, we select checkpoints at epochs $\{50, 100, 150, 200\}$.

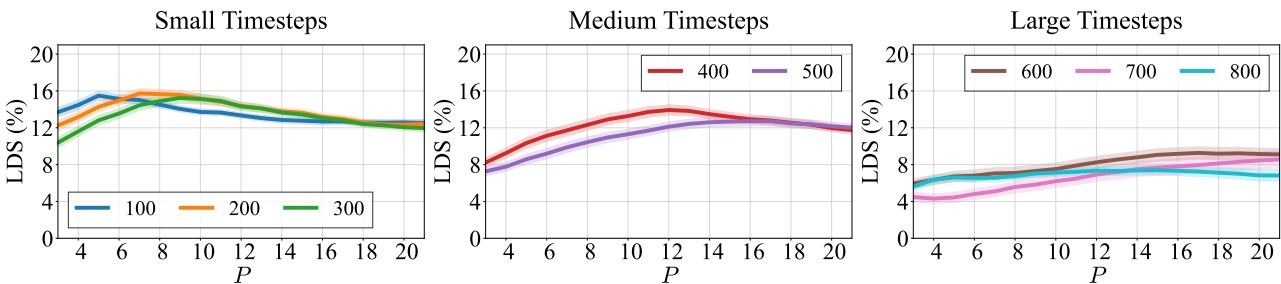

*Figure 9.* LDS (%) on the CelebA validation set across different patch sizes and timesteps at the original resolution. **Left**: small timesteps ($t\leq300$), where moderate patch sizes achieve the highest scores. **Middle**: medium timesteps ($t=400, 500$), where larger patches outperform smaller ones due to higher noise. **Right**: large timesteps, where noise dominates, and the informative signal is minimal.

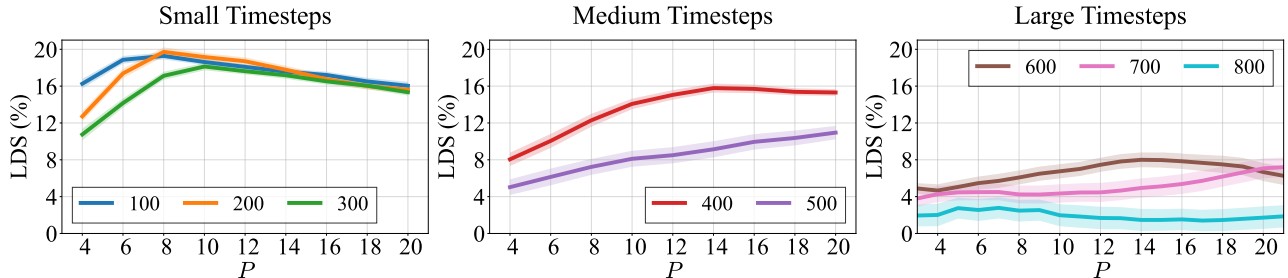

*Figure 10.* LDS (%) on the CIFAR-2 validation set across different patch sizes and timesteps at the low-resolution scale. The left panel shows small forward timesteps ($t = 100, 200, 300$), where moderate patch sizes yield the highest scores. The middle panel shows medium timesteps ($t = 400, 500$), where larger patches perform better. The right panel shows large timesteps ($t = 600, 700, 800$), where noise dominates, and the informative signal is limited.

**GAS.** This is a "renormalized" version of TracInCP that leverages cosine similarity in place of raw dot products (Hammoudeh & Lowd, 2022).

**TRAK.** As discussed in (Zheng et al., 2024), TRAK (Park et al., 2023a) can be extended to the diffusion setting, and the retraining-free TRAK can be implemented as:

$$\Phi_{\text{TRAK}} = \left[\phi\left(\boldsymbol{x}^1\right), \cdots, \phi\left(\boldsymbol{x}^N\right)\right]^\top, \text{where } \phi(\boldsymbol{x}) = \mathcal{P}^\top \nabla_\theta \mathcal{L}_{\text{Simple}}(\boldsymbol{x}, \theta^*)$$

$$\tau(\boldsymbol{x}, \boldsymbol{z}^{(n)}; \mathbb{S}) = \left(\mathcal{P}^\top \nabla_\theta \mathcal{L}_{\text{Simple}}(\boldsymbol{x}, \theta^*)\right)^\top \cdot \left(\Phi_{\text{TRAK}}^\top \Phi_{\text{TRAK}} + \lambda I\right)^{-1} \cdot \mathcal{P}^\top \nabla_\theta \mathcal{L}_{\text{Simple}}\left(\boldsymbol{z}^{(n)}, \theta^*\right).$$

where $\lambda I$ is included for numerical stability and regularization.

**D-TRAK.** Similar to TRAK,

$$\Phi_{\text{D-TRAK}} = \left[\phi\left(\boldsymbol{x}^1\right), \cdots, \phi\left(\boldsymbol{x}^N\right)\right]^\top, \text{where } \phi(\boldsymbol{x}) = \mathcal{P}^\top \nabla_\theta \mathcal{L}_{\text{Square}}(\boldsymbol{x}, \theta^*)$$

$$\tau(\boldsymbol{x}, \boldsymbol{z}^{(n)}; \mathbb{S}) = \left(\mathcal{P}^\top \nabla_\theta \mathcal{L}_{\text{Square}}(\boldsymbol{x}, \theta^*)\right)^\top \cdot \left(\Phi_{\text{D-TRAK}}^\top \Phi_{\text{D-TRAK}} + \lambda I\right)^{-1} \cdot \mathcal{P}^\top \nabla_\theta \mathcal{L}_{\text{Square}}\left(\boldsymbol{z}^{(n)}, \theta^*\right).$$

where $\lambda I$ is also included for numerical stability and regularization, as in TRAK. Here $\mathcal{L}_{\text{Square}} = \|\epsilon_\theta(\boldsymbol{x}_t, t)\|^2$. In D-TRAK, $\mathcal{L}_{\text{Square}}$ is used to construct both the feature map and the query/training-side gradients in the attribution score. Additionally, the output function $\mathcal{L}_{\text{Square}}$ could be replaced by other functions.

## C. Ablation Studies

**Patch size selection.** For completeness, we also study the effect of patch size by evaluating LDS across $P \in [3, 21]$ at different timesteps on CelebA. To isolate timestep effects, we fix $\mathcal{T}$ to a single $t$ and vary $P$. Fig. 9 shows LDS as a function of $P$ for each $t$. The trend mirrors CIFAR-2: the optimal patch size generally grows with $t$. At small forward timesteps ($t \leq 300$), small to moderate patches (e.g., $P = 5, 7, 9$) yield the highest scores, indicating local patterns dominate when noise is low. At mid-range timesteps ($400 \leq t \leq 500$), larger patches perform better (peaks shift to $P = 11, 13$), likely due

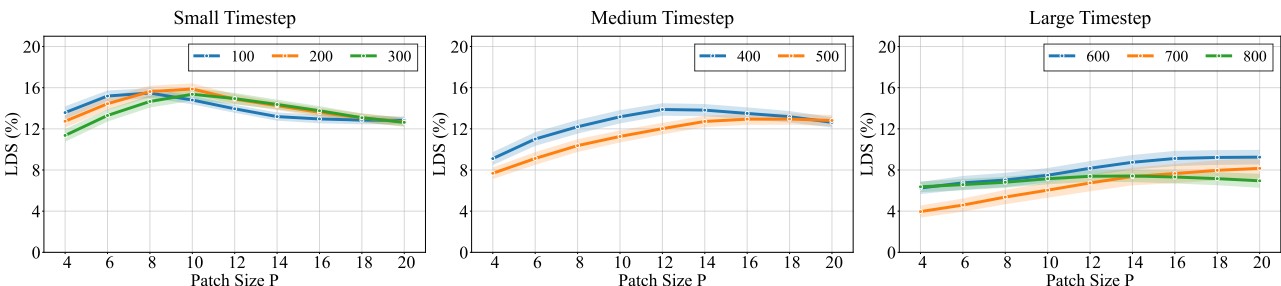

*Figure 11.* LDS (%) on the CelebA validation set across different patch sizes and timesteps at the low resolution. **Left**: small timesteps ($t \leq 300$), where moderate patch sizes achieve the highest scores. **Middle**: medium timesteps ($t = 400, 500$), where larger patches outperform smaller ones due to higher noise. **Right**: large timesteps, where noise dominates, and the informative signal is minimal.

to the need for more contextual information under higher noise. In the high-noise regime ($t \geq 600$), LDS drops and curves flatten across all patch sizes, suggesting the informative signal is limited.

**Multiscale influence.** We further compare the optimal patch size between the original and the low-resolution settings on CIFAR-2. As shown in Fig. 5 and Fig. 10, at the small forward timestep $t = 100$, the optima differ notably: the original-resolution curve peaks at $P = 5$, whereas the low-resolution curve peaks at $P = 8$, indicating that coarser inputs benefit from slightly larger spatial context. By contrast, at $t = 200$ and $t = 300$, the two resolutions behave very similarly: the original-resolution curve peaks at $P = 7$ and $P = 9$, while the low-resolution curve peaks at $P = 8$ and $P = 10$, with near-overlapping curves around their maxima. At mid timesteps ($t = 400, 500$), both resolutions favor large patch sizes, with $P = 21$. These results show that downscaling mainly shifts the optimal patch size toward larger values at small timesteps, while mid/large timesteps show qualitatively similar trends across resolutions.

# D. Additional Experiment Results

## D.1. Generalization Across Architectures and Training Objectives

To further examine the robustness of NDA across different model designs and training paradigms, we extend our evaluation along two orthogonal dimensions: *model architecture* and *training objective*. Specifically, we replace the original U-Net backbone with a ViT-based architecture (UViT), and investigate both DDPM and Flow Matching objectives under this new backbone.

**UViT-DDPM.** We first replace the original U-Net backbone with a ViT-based architecture, adopting UViT (Bao et al., 2023) as the base model. The model is still trained using the standard DDPM objective $\mathcal{L}_{\text{Simple}}$ in Eq. (3), while keeping the rest of the DDPM training protocol unchanged.

**UViT-Flow Matching.** We further consider a more substantial modification by combining the UViT architecture with a Flow Matching training objective (Lipman et al., 2023). We implement Flow Matching using a Rectified Flow formulation (Liu et al., 2023), where the model learns a velocity field along a linear interpolation between data and noise. Specifically, given a data sample $\boldsymbol{x}_0$ and Gaussian noise $\boldsymbol{\epsilon} \sim \mathcal{N}(0, I)$, we sample $t \sim \mathcal{U}(0, 1)$ and construct

$$\boldsymbol{x}_t = (1 - t)\boldsymbol{x}_0 + t\boldsymbol{\epsilon},$$

We then train the network to predict the target velocity

$$\mathbf{v}^* = \boldsymbol{\epsilon} - \boldsymbol{x}_0$$

by minimizing the following Flow Matching objective:

$$\mathcal{L}_{\text{FM}}(\theta) = \mathbb{E}_{\boldsymbol{x}_0, \boldsymbol{\epsilon}, t} \left[ \|\mathbf{v}_\theta(\boldsymbol{x}_t, t) - (\boldsymbol{\epsilon} - \boldsymbol{x}_0)\|_2^2 \right]. \tag{16}$$

This setting changes both the model architecture and the training objective, allowing us to evaluate NDA under a different training paradigm.

*Table 3.* LDS (%) on CIFAR-2, CIFAR-10, and CelebA validation datasets for UViT target models under DDPM and Flow Matching objectives.

| Setting    (Dataset) | Raw pixel (cos.) | CLIP (cos.) | NDA | D-TRAK |
|---|---|---|---|---|
| UViT–DDPM (CIFAR-2) | 10.81 | 4.32 | 17.13 | 21.17 |
| UViT–DDPM (CIFAR-10) | 1.73 | 5.13 | 6.07 | 6.62 |
| UViT–DDPM (CelebA) | 8.84 | 6.92 | 11.62 | 15.18 |
| UViT-Flow Matching (CIFAR-2) | 3.95 | 5.30 | 15.61 | 15.20 |
| UViT-Flow Matching (CIFAR-10) | 7.16 | 1.51 | 15.53 | 16.61 |
| UViT-Flow Matching (CelebA) | 3.85 | 3.28 | 8.11 | 9.73 |

**UViT architecture.**    For all UViT-based models, we adopt the UViT-Small configuration from (Bao et al., 2023), which has approximately $44.4$M parameters. While larger UViT variants can improve generation quality under the DDPM objective on CelebA, we observe that such scaling does not consistently benefit the Flow Matching objective. Therefore, we use the UViT-Small backbone across both objectives to ensure a consistent and fair evaluation of attribution methods.

**Training details (Flow Matching).**    For the UViT-Flow Matching model, we employ the AdamW optimizer with learning rate $2 \times 10^{-4}$, weight decay $0.03$, and $\beta = (0.99, 0.99)$, using a batch size of 128. We train for a fixed number of optimization steps, using 200K steps on CIFAR-10 and 50K steps on CIFAR-2 and CelebA. A cosine learning rate schedule with 5K warmup steps is adopted, and an exponential moving average (EMA) of the model parameters is maintained for stable sampling. All hyperparameters are shared across datasets and are not specifically tuned for attribution evaluation.

Table 3 reports LDS results for NDA and the baselines under the **UViT-DDPM** and **UViT-Flow Matching** settings described above. NDA consistently outperforms the Raw pixel and CLIP baselines across these settings and achieves competitive LDS relative to the gradient-based D-TRAK reference in several cases. These results suggest that the attribution signal captured by NDA remains useful under the tested changes in model architecture and training objective.

### D.2. Patch-Based and Image-Level Baselines

#### D.2.1. NAIVE PATCH SIMILARITY ACROSS TIMESTEPS

To test whether NDA's gains can be explained by naive patch similarities across timesteps, we implement two naive patch-wise influence baselines (cf. Eq. (11)) that directly measure patch-level distance across timesteps, without mirroring the score function:

1. **Patch L2 similarity:**
$$\tau(\boldsymbol{x}_{t,\Omega_\ell}, \boldsymbol{u}) = - \left\| \boldsymbol{x}_{t,\Omega_\ell} - \boldsymbol{u} \right\|^2.$$

2. **Noise-weighted patch L2 similarity:**
$$\tau(\boldsymbol{x}_{t,\Omega_\ell}, \boldsymbol{u}) = - \frac{\left\| \boldsymbol{x}_{t,\Omega_\ell} - \sqrt{\bar{\alpha}_t}\, \boldsymbol{u} \right\|^2}{2(1 - \bar{\alpha}_t)}.$$

We keep the same aggregation procedure from patch-level influence scores to image-level attribution (i.e., Eq. (12)), and report results averaged across timesteps $\mathcal{T} = \{100, 200, 300, 400, 500\}$ in Table 4. As shown, naive patch similarity baselines yield notably inferior performance compared to NDA across all datasets.

#### D.2.2. PATCH-WISE BASELINES

We include results on three patch-wise baselines. For all baselines, we extract local patches from the generated image $\boldsymbol{x}$ and each training image $\boldsymbol{z}$ using the same setup as in Sec. 3.1. Given these patches:

1. **Patch-wise Raw pixel**: We compute cosine similarity between every patch of $\boldsymbol{x}$ and every patch of $\boldsymbol{z}$. For each patch of $\boldsymbol{x}$, we select its top-$k$ most similar patches in $\boldsymbol{z}$ and sum their scores, then aggregate these per-patch sums over all patches of $\boldsymbol{x}$ to obtain an image-level similarity score.

*Table 4.* LDS (%) comparison between naive patch similarity baselines across timesteps and NDA on CIFAR-2, CIFAR-10, and CelebA.

| Method | CIFAR-2 | | CIFAR-10 | | CelebA | |
|---|---|---|---|---|---|---|
| | Validation | Generation | Validation | Generation | Validation | Generation |
| Patch L2 similarity | 5.94 | 3.91 | 1.60 | 1.67 | 7.81 | 7.07 |
| Noise-weighted patch L2 similarity | 5.34 | 3.15 | 1.53 | 1.82 | 7.43 | 6.58 |
| NDA | 24.88 | 15.91 | 11.81 | 7.41 | 16.89 | 13.92 |

*Table 5.* LDS (%) comparison between patch-wise Raw pixel / CLIP / DINOv2 baselines and NDA on CIFAR-2, CIFAR-10, and CelebA.

| Method | CIFAR-2 | | CIFAR-10 | | CelebA | |
|---|---|---|---|---|---|---|
| | Validation | Generation | Validation | Generation | Validation | Generation |
| Patch-wise Raw pixel | 4.16 | 2.42 | 1.18 | 2.03 | 3.26 | 3.77 |
| Patch-wise CLIP | 6.61 | 4.73 | 1.20 | 1.61 | 6.35 | 5.76 |
| Patch-wise DINOv2 | 14.28 | 5.42 | 7.77 | 3.31 | 10.98 | 3.58 |
| NDA | **24.88** | **15.91** | **11.81** | **7.41** | **16.89** | **13.92** |

2. **Patch-wise CLIP**: We encode each patch with CLIP, and then compute cosine similarities between patch embeddings of $x$ and $z$, and apply the same procedure as above to get an image-level score.

3. **Patch-wise DINOv2**: We encode each patch with a pre-trained DINOv2 (Oquab et al., 2024) feature extractor, and compute cosine similarities between patch embeddings of $x$ and $z$. We then apply the same top-$k$ aggregation procedure as above to obtain an image-level score.

We tune the patch size $P \in [3, 21]$ and the top-$k$ parameter separately for each patch-wise baseline on the CIFAR-2 validation dataset. For the patch-wise Raw pixel baseline, $P = 9$ and $k = 100$ perform best; for the patch-wise CLIP baseline, $P = 5$ and $k = 100$ work best; for the patch-wise DINOv2 baseline, $P = 7$ and $k = 100$ work best. We fix these choices for all datasets and report the corresponding results in Table 5.

As shown in Table 5, NDA outperforms all three patch-wise baselines (Raw pixel, CLIP, DINOv2) on every dataset, suggesting that simply patchifying the images does not improve LDS and that NDA's gains arise from insights theoretically motivated by the analytical score, rather than from patchification alone.

### D.2.3. IMAGE-WISE NDA BASELINE

In addition to patch-wise attribution, we consider an Image-wise NDA baseline that applies the optimal score formulation in Eq. (5) directly at the full-image level. Specifically, for each timestep $t \in \mathcal{T}$, we compute weights $W_t(z^{(n)}|x_t)$ according to Eq. (6) and use them as image-level scores without any patch decomposition or spatial aggregation. All other settings, including the noise schedule, the selected timestep set $\mathcal{T}$, and the evaluation protocol, are kept identical to those used for patch-wise NDA. We report LDS results on CIFAR-2, CIFAR-10, and CelebA in Table 6. As shown in the table, the image-wise NDA baseline is clearly weaker than our NDA method. This is expected, as the ideal score in Eq. (5) does not induce any generalization beyond the training set, while NDA is inspired by the analytical score under locality and equivariance assumptions, which provide the necessary inductive bias for stronger attribution.

## E. Runtime Analysis

We analyze the computational cost of our NDA method and compare it with a representative gradient-based attribution method, D-TRAK. Algorithmically, for a batch of $B$ test images, $|T|$ timesteps, and $N$ training images of resolution $H \times W$, the time complexity of our NDA implementation is $\mathcal{O}(B|T|N(HW)^2 P^2)$, i.e., it scales linearly in $B$, $|T|$, $N$, and the patch area $P^2$, and quadratically in the number of spatial locations $HW$. For D-TRAK, the attribution stage (reported as D-TRAK Attr. in Table 7) consists of gradient computation across timesteps, followed by a projection step, with cost $\mathcal{O}(|T| \cdot \text{GRAD\_TIME} + \text{PROJ\_TIME})$. We provide wall-clock runtimes for attribution computation on a single NVIDIA A100 GPU using our PyTorch implementation. All runtimes are measured on the validation sets of CIFAR-2, CIFAR-10, and

Table 6. LDS (%) comparison between image-wise NDA baseline and NDA on CIFAR-2, CIFAR-10, and CelebA.

| Method | CIFAR-2 | | CIFAR-10 | | CelebA | |
|---|---|---|---|---|---|---|
| | Validation | Generation | Validation | Generation | Validation | Generation |
| Image-wise NDA | 9.96 | 5.01 | 6.07 | 2.27 | 5.82 | 3.93 |
| NDA | 24.88 | 15.91 | 11.81 | 7.41 | 16.89 | 13.92 |

Table 7. Wall-clock runtimes of NDA and D-TRAK on a single NVIDIA A100 GPU using our PyTorch implementations.

| Dataset | #Samples | D-TRAK Train (hh:mm) | D-TRAK Attr. (hh:mm) | NDA (hh:mm) |
|---|---|---|---|---|
| CIFAR-2 | 5,000 | 00:29 | 00:44 | 0:50 |
| CIFAR-10 | 50,000 | 03:35 | 06:20 | 8:58 |
| CelebA | 5,000 | 00:53 | 04:36 | 9:12 |

CelebA. From a computational perspective, NDA admits a high degree of parallelism: it does not involve deep hierarchical structures, and all patches can be computed simultaneously, providing significant potential for acceleration.

## F. Stride Study on CelebA

To support the stride-2 choice used for ArtBench-2 ($256 \times 256$) in the main paper, we report a stride study on CelebA. Table 8 compares NDA at stride-1 and stride-2 against D-TRAK on the same target model and evaluation protocol. Stride-$s$ subsampling extracts patches at every $s$-th spatial location during patch extraction, while keeping Eq. (11) and the image-level aggregation (Eq. 15) unchanged.

On CelebA, stride-2 reduces runtime from 9:12 to 1:16 with negligible LDS change ($16.89 \rightarrow 17.20$ on validation, $13.92 \rightarrow 13.73$ on generation), and is substantially faster than D-TRAK attribution (04:36). Since neighboring patches are often redundant at higher resolution, this result motivates the stride-2 choice for ArtBench-2 in Sec. 4.2.

## G. Visualization

We provide additional visualizations, including counterfactual examples and proponent–opponent analyses.

### G.1. Counterfactual Visualization

We include more counterfactual visualizations in Fig. 12 and 13. As shown, NDA often identifies training examples whose removal produces a marked change in the output of a model retrained with the same seed. In Fig. 12 (right column, third row), models retrained after removing images selected by Random and CLIP still synthesize an "automobile". In contrast, removing images selected by NDA or D-TRAK yields an image that resembles a mixture of "automobile" and "horse". Notably, the image after NDA's removal shows clearer horse morphology (e.g., head and leg outlines) than that after D-TRAK. These qualitative examples provide additional support that NDA can identify training images with substantial influence on a given target.

### G.2. Proponent–Opponent Visualization

Following Pruthi et al. (2020), we use the terms proponents and opponents for training examples with positive and negative influence scores, respectively. In our figures, this interpretation applies directly to D-TRAK: positive D-TRAK scores are shown on the proponent side, while negative scores are shown on the opponent side. For CLIP and NDA, we use the same left–right layout only to visualize rankings, where the two sides correspond to the highest- and lowest-scoring examples under cosine similarity and our attribution score, respectively. For each target, we retrieve and visualize the top-5 high-scoring examples and the top-3 low-scoring examples. Qualitative results on CIFAR-2, CIFAR-10, and CelebA are shown in Figs. 14, 15, and 16.

*Table 8.* Stride study on CelebA. Stride-2 preserves LDS while reducing runtime by $\sim 7\times$, and is faster than D-TRAK attribution.

| Setting | Val LDS (%) | Gen LDS (%) | Runtime (hh:mm) |
|---|---|---|---|
| NDA (stride=1) | 16.89 | 13.92 | 9:12 |
| NDA (stride=2) | 17.20 | 13.73 | **1:16** |
| D-TRAK | 22.83 | 16.84 | 04:36 |

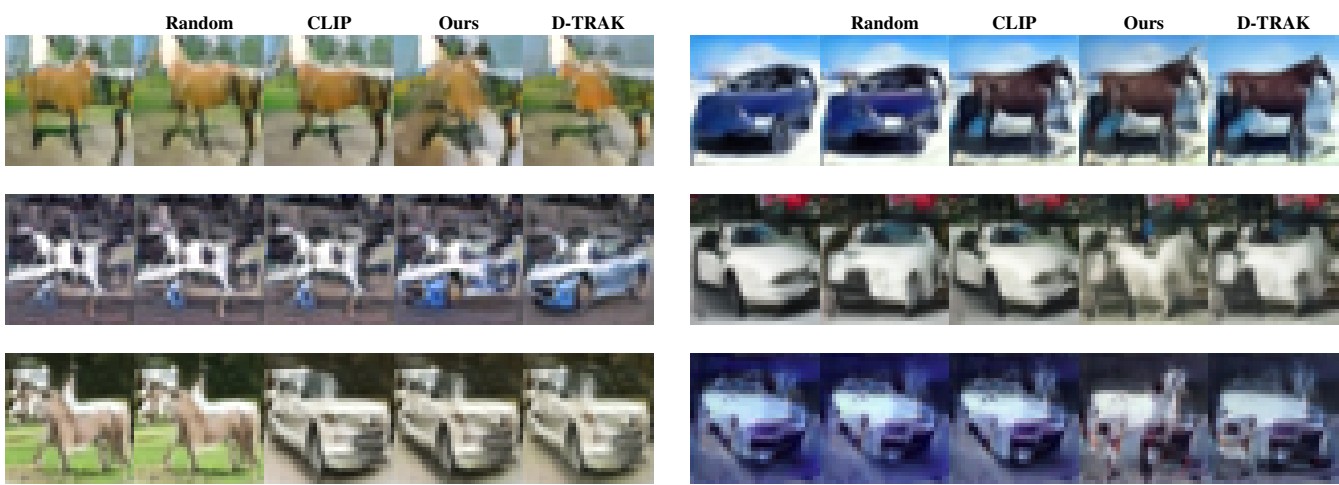

*Figure 12.* Counterfactual visualization on CIFAR-2 dataset. We compare samples generated by retrained models with different attribution methods using the same seed.

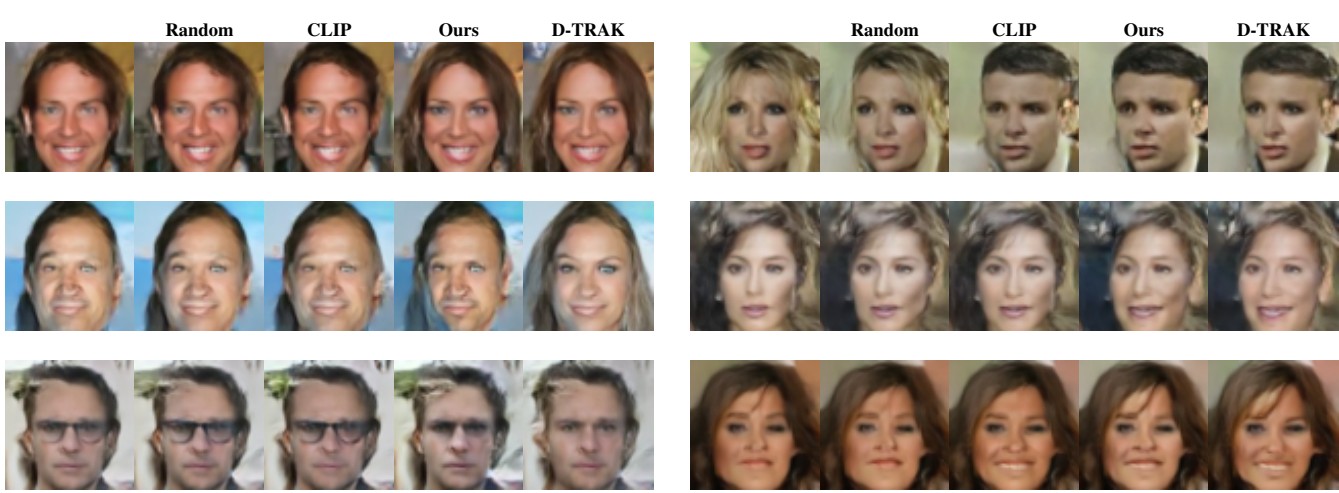

*Figure 13.* Counterfactual visualization on the CelebA dataset. We compare samples generated by retrained models with different attribution methods using the same seed.

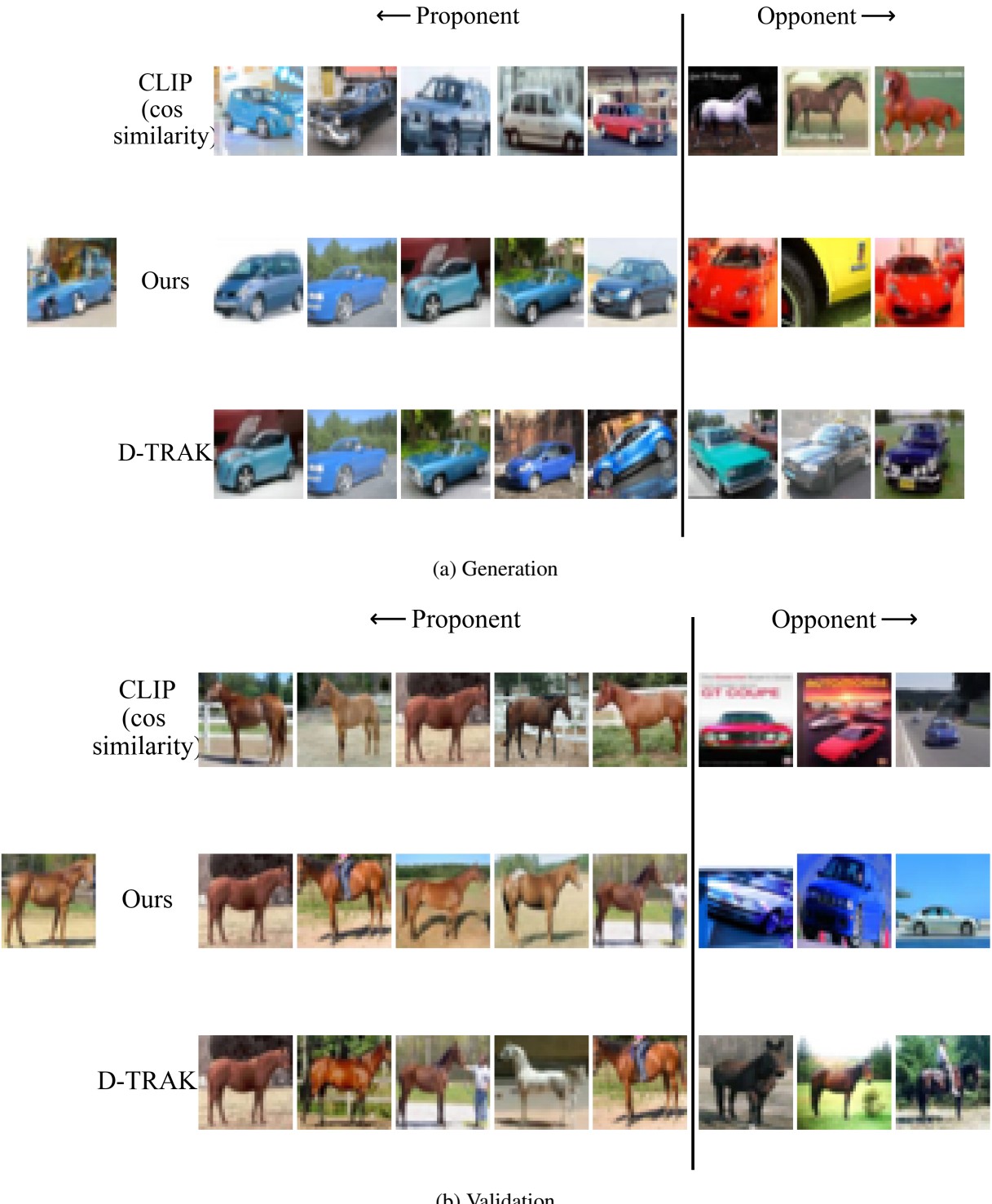

(a) Generation

(b) Validation

Figure 14. Proponents and opponents visualization on the CIFAR-2 dataset using CLIP, NDA, and D-TRAK, with attribution scores averaged over timesteps. For each sample of interest, we retrieve training samples based on attribution scores. The left column shows the 5 highest-scoring training samples, and the right column shows the 3 lowest-scoring training samples.

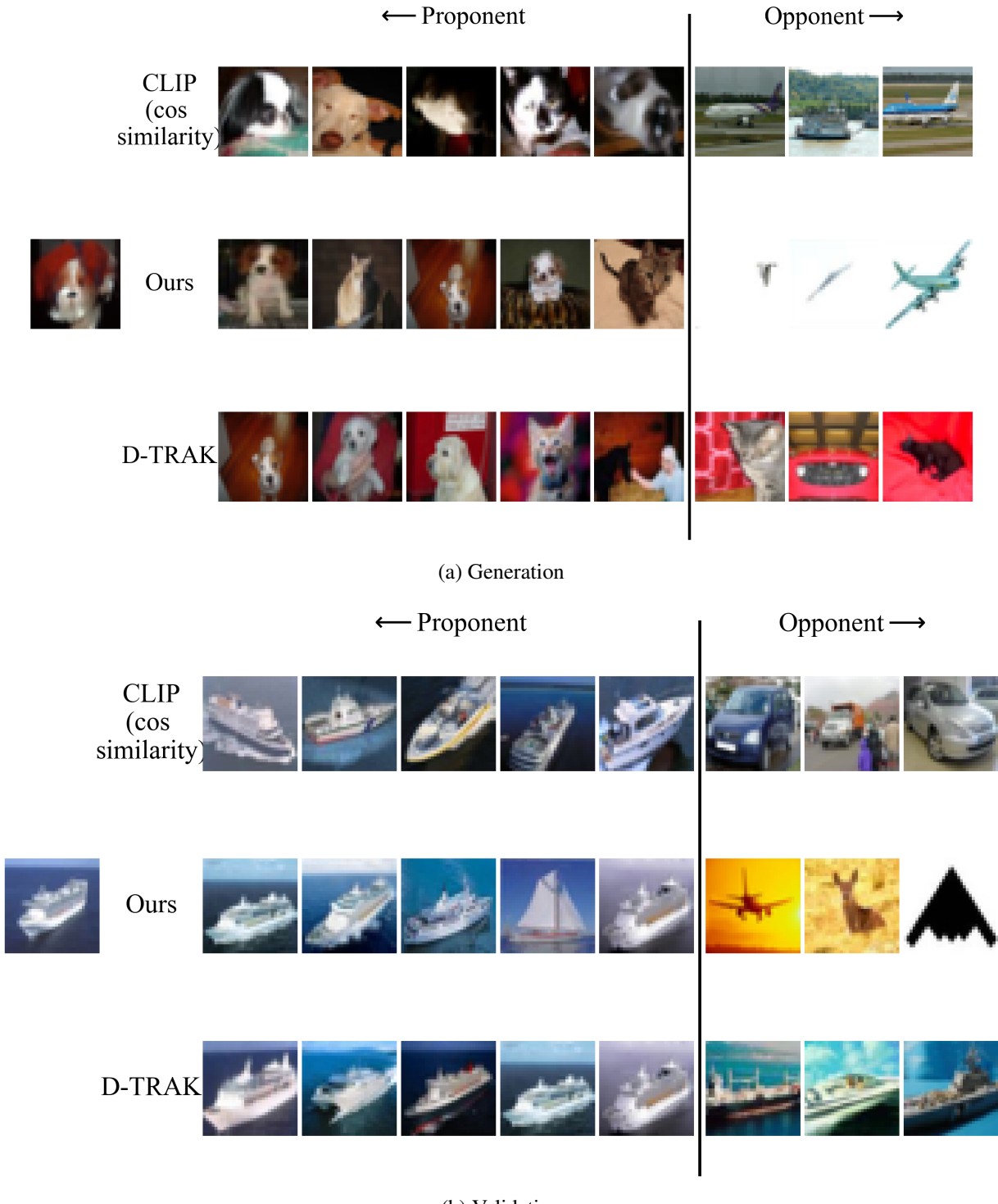

(a) Generation

(b) Validation

*Figure 15.* Proponents and opponents visualization on the CIFAR-10 dataset using CLIP, NDA, and D-TRAK, with attribution scores averaged over timesteps. For each sample of interest, we retrieve training samples based on attribution scores. The left column shows the 5 highest-scoring training samples, and the right column shows the 3 lowest-scoring training samples.

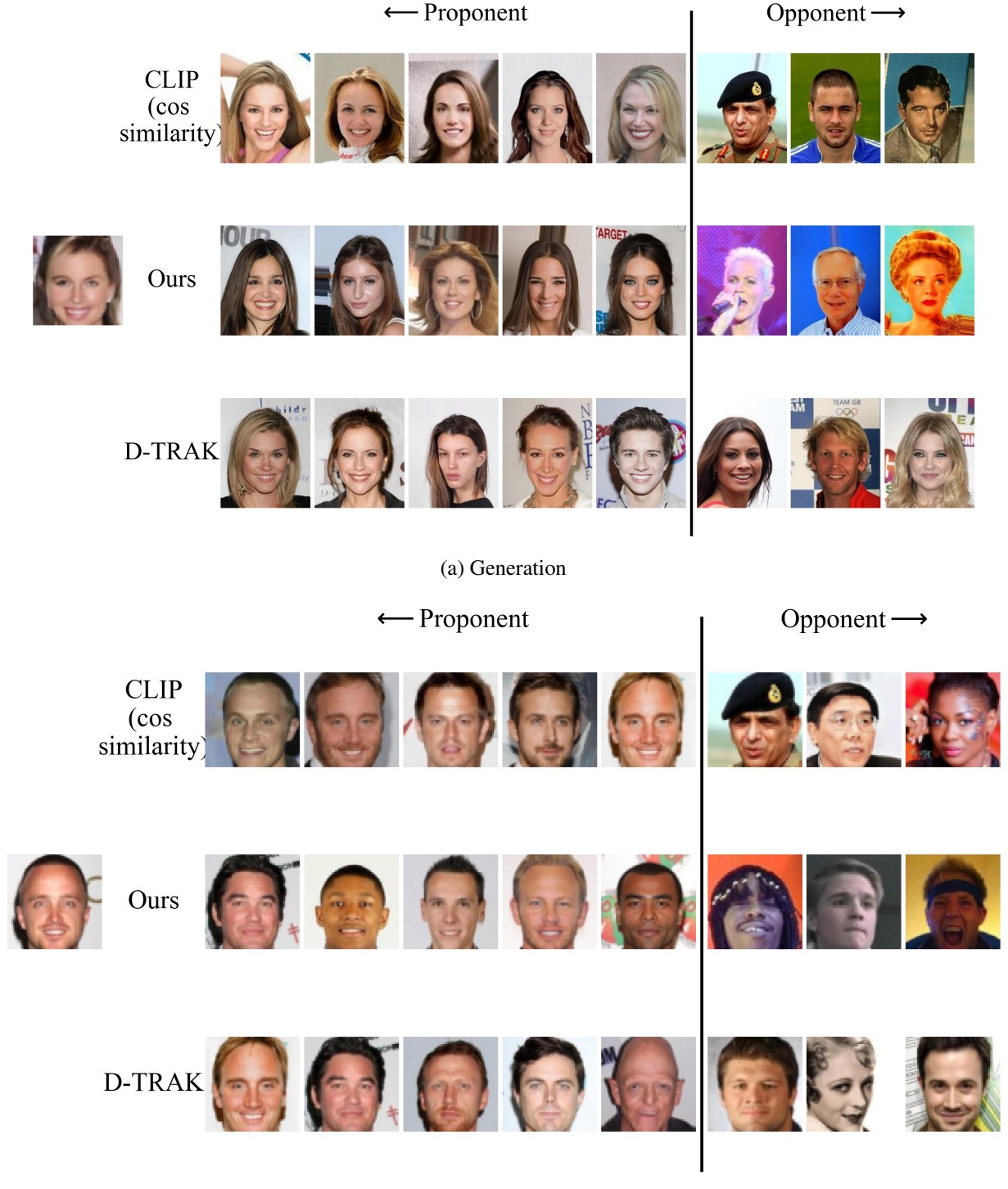

(a) Generation

(b) Validation

*Figure 16.* Proponents and opponents visualization on the CelebA dataset using CLIP, NDA, and D-TRAK, with attribution scores averaged over timesteps. For each sample of interest, we retrieve training samples based on attribution scores. The left column shows the 5 highest-scoring training samples, and the right column shows the 3 lowest-scoring training samples.

