# OpenReview forum: "Nonparametric Data Attribution for Diffusion Models"
_ICML.cc/2026/Conference — ICML 2026 regular_

### Official Review · Reviewer_9tdz · 2026-03-07

**Soundness:** 2
**Presentation:** 3
**Significance:** 2
**Originality:** 3
**Overall Recommendation:** 4
**Confidence:** 4

**Summary:**

This paper presents a model-free data attribution method for diffusion models. The key insight is to leverage Kamb and Ganguli (2024)'s characterization of the optimal score function for a finite training set under equivariance and locality constraints. The authors define an attribution score that measures the relative similarity of each patch in a generated sample to patches in the training set and aggregate these scores across patches and noise levels. This score is non-parametric in the sense that it does not use information from the trained diffusion model but instead directly queries the training set. The authors show that their method achieves competitive linear datamodeling scores relative to gradient-based methods like TRAK and D-TRAK and also performs competitively at identifying influential training samples. In addition to these key empirical results, the authors also ablate the impact of several key hyperparameters and visualize the most influential training samples for several model samples.

**Compliance With Llm Reviewing Policy:**

Affirmed.

**Final Justification:**

In their rebuttal, the authors included results from a stride study, which shows that a simple patch subsampling scheme significantly reduces their method's wall-clock time while mostly maintaining its performance. In light of these results, I have raised my score to a weak accept. However, I maintain my concerns about their method's performance relative to D-TRAK and its theoretical connection to diffusion models, which prevent me from recommending acceptance without reservations.

**Key Questions For Authors:**

1. Computing the proposed attribution score seems to incorporate lots of redundancy across patches in the model sample. In particular, Eq (12) includes a sum over all pixel locations in the model sample. Wouldn't we intuitively expect patches centering at neighboring pixels to be nearly identical and to therefore have very similar similarity scores with respect to the training patches? Can we leverage this to subsample patches in the model sample and perhaps lighten the computational burden?

2. It seems like the patch-wise influence scores (11) are dominated by the nearest patches to $x_{t, \Omega_\ell}$ in the training set, especially when $1 - \bar{\alpha}_t$ is close to 0. Could we use this property to estimate the influence scores via approximate nearest neighbor queries? Works like "Closed-Form Diffusion Models" by Scarvelis et al. (2025) use similar techniques to estimate the optimal score for a training set.

3. I'm not sure how much scientific value there is in the experiment from Section 4.4. Does it show something beyond the fact that training images with large NDA scores tend to contain patches that are similar to patches in the model sample? Isn't this tautological? It seems like this is baked into the definition of the image-level attribution score.

**Limitations:**

Yes

**Strengths And Weaknesses:**

This is a well-written paper that presents a novel and reasonably well-motivated data attribution method (NDA) for diffusion models. (As the authors allude to in Section 3.5, it can in principle be used for other classes of generative models like CNFs, though the theoretical motivation would be perhaps weaker for such models.) My main critique of this method is that I'm not sure if it works well enough in practice to justify the great expense of iterating over the entire training set to compute attribution scores. In particular, the results in Table 1 indicate that the proposed method underperforms D-TRAK, a SOTA gradient-based method for data attribution -- often by a significant margin. While I could buy that this is a reasonable price to pay for greater efficiency, the runtime analysis in Table 6 (Appendix E) shows that the proposed NDA method is not efficient enough to justify worse performance than D-TRAK. In particular, while NDA is faster than D-TRAK's total runtime on small datasets like CIFAR, it takes ~67% *longer* to run on CelebA. This gap seems to be a function of the dimension of the underlying data, and I worry that it will expand further when one attempts to apply NDA to high-resolution images.

More broadly, I'm not sure if the model-free nature of this data attribution method is a feature rather than a bug. The authors attempt to address this point in Section 3.5, but I'm a bit skeptical of their arguments. While this method is inspired by Kamb and Ganguli's results on the structure of optimal score models under certain constraints, the theoretical relationship between the NDA score and diffusion models is somewhat weak, and it seems like NDA scores are more like a similarity measure between images than a data attribution method for diffusion models. Moreover, whereas the authors motivate their method relative to gradient-based methods by stating that "in scenarios where where users seek copyright protection against infringement by proprietary models, the model gradients may not always be accessible," it seems implausible that users would have access to the proprietary models' training sets instead.

---

> ### Author Rebuttal · Authors · 2026-03-31
>
> Thank you for your valuable review and suggestions. Below we respond to the comments in Weaknesses (W) and Questions (Q).
>
> ---
>
> ***W1: Are NDA's gains explained by generic similarity rather than diffusion-specific attribution?***
>
> Thank you for raising this concern. We tested this directly with a range of generic similarity baselines at both the patch and image level. In the table below, we compare patch L2, noise-weighted patch L2, and an image-wise NDA variant, all of which are substantially weaker than NDA:
>
> | Method | CIFAR-2 Val | CIFAR-10 Val | CelebA Val |
> |--------|-------------|--------------|------------|
> | Patch L2 similarity | 5.94 | 1.60 | 7.81 |
> | Noise-weighted patch L2 | 5.34 | 1.53 | 7.43 |
> | Image-wise NDA | 9.96 | 6.07 | 5.82 |
> | NDA (ours) | **24.88** | **11.81** | **16.89** |
>
> DINOv2 reaches 14.28 / 5.42 on CIFAR-2 (Val / Gen), versus 24.88 / 15.91 for NDA. Combined with the patch-wise Raw pixel / CLIP and image-wise baselines in App. D.2, Tables 3--5, this indicates that NDA's gains are not explained by simple patchification or generic image-/patch-level similarity, even when stronger retrieval features are used.
>
> ---
>
> ***W2 & Q2: Higher-resolution evaluation, patch subsampling, and nearest-patch approximation.***
>
> Thank you for raising this concern. Our target use case is not a strict black-box API with no data access; rather, it is the setting where the candidate training dataset is available or auditable, but the deployed generator is proprietary or otherwise inaccessible. As discussed in **R1-W2** above, we added a CelebA stride study to directly revisit the wall-clock issue. The stride=2 variant is still the same NDA score: Eq. (11) and the image-level aggregation are unchanged, and we simply evaluate them on a sparser query-patch grid. In that setting, runtime drops from 9h 12m to 1h 16m while LDS changes little (16.89 / 13.92 $\rightarrow$ 17.20 / 13.73 on Val / Gen), so this substantially alleviates the CelebA cost concern rather than requiring a different attribution rule. We address the higher-resolution setting in R1-W2 on ArtBench. Here we focus on the reviewer’s CelebA wall-clock concern and show that simple patch subsampling already reduces runtime substantially with little LDS change.
>
> For the reviewer's nearest-patch suggestion, we additionally tested exact patch-level oracle top-M truncation on CIFAR-10 validation: for each query patch, we keep the exact top-M training patches in Eq. (11), renormalize over them, and then apply the same image-level aggregation as full NDA. Full NDA gives **11.81** LDS, while **M=2048** and **M=8192** give **11.12** and **11.34**. We present this conservatively as evidence that a moderate patch shortlist retains much of the signal, not that ANN is solved or that exact full normalization is unnecessary.
>
> ---
>
>
>
>
>
> ***Q1: Are dense neighboring patches really needed?***
>
> Thank you for this question. As discussed in **R1-W2** above, the stride study shows that we do not need to extract patches centered at all pixels in order to retain high LDS in practice. At the same time, Image-wise NDA is clearly weaker than patch-wise NDA (CIFAR-2 Val: 9.96 vs. 24.88; CIFAR-10 Val: 6.07 vs. 11.81; CelebA Val: 5.82 vs. 16.89). So patch-level attribution matters, but we can do some patch subsampling in practice.
>
> ---
>
> ***Broader concern: Is NDA just similarity rather than attribution?***
>
> Thank you for highlighting this concern. This is exactly why we added the baselines above and in **App. D.2, Tables 3--5**. Naive patch similarity, patch-wise Raw pixel / CLIP, stronger retrieval features such as DINOv2, and Image-wise NDA are all substantially weaker than NDA. In the shared 100-image counterfactual study, NDA also gives the largest $\ell_2$ change at 1\%, 5\%, and 10\% removal. Taken together, these results are harder to explain as generic visual similarity alone and instead support the diffusion-specific weighting and normalization as the key ingredient.
>
> ---
>
> ***Q3: Is Section 4.4 tautological?***
>
> Thank you for raising this point. We agree that the role of Section 4.4 should be clarified. It is not intended as the main faithfulness evidence; that role is played by LDS and the counterfactual experiment. Section 4.4 serves a different purpose: it visualizes which patches in the top-attributed training images align with influential regions of the test image, making the attribution spatially interpretable. That type of patch-level interpretation is not captured by a scalar LDS score or by image-level attribution alone, so we present Section 4.4 as interpretability evidence rather than as a separate faithfulness claim.

---

> > ### Author Rebuttal · Reviewer_9tdz · 2026-04-03
> >
> > Thank you for your thorough rebuttal and for the additional experiments. In light of the stride study, which shows that a simple patch subsampling scheme significantly reduces their method's wall-clock time while maintaining its performance, I am prepared to raise my score to a weak accept. However, I maintain my concerns about their method's performance relative to D-TRAK and its theoretical connection to diffusion models.

---

> > > ### Author Response · Authors · 2026-04-07
> > >
> > > Thank you for the follow-up and for the updated assessment. We appreciate that the stride study helped clarify the wall-clock concern. On the D-TRAK point, our intended comparison is within the setting where target-model gradients are unavailable, while D-TRAK remains stronger when such gradients are available. On the theory side, the connection we intend to claim is that NDA is motivated by the local weighting term in the diffusion optimal-score analysis. After the rebuttal period, we would refine the wording in the abstract, introduction, and discussion to make these two points more explicit.

---

### Official Review · Reviewer_zchd · 2026-03-11

**Soundness:** 3
**Presentation:** 3
**Significance:** 4
**Originality:** 4
**Overall Recommendation:** 5
**Confidence:** 3

**Summary:**

This paper introduces a nonparametric framework for data attribution in diffusion models. Unlike existing "parametric" methods (e.g., D-TRAK) that require internal gradient access or "leave-one-out" methods that require expensive retraining, the proposed method is derived from the optimal score function. By treating the score as a weighted average of training samples, the authors propose a multiscale, patch-level similarity metric to quantify the influence of training data on generated outputs. The method is computationally efficient and provides spatial interpretability through heatmaps.

**Compliance With Llm Reviewing Policy:**

Affirmed.

**Final Justification:**

I kept my "Accept" score because the method proposed is simple yet effective.

**Key Questions For Authors:**

1. Scalability (Many-to-Many): The current method is presented as a "one-to-many" check (one generation vs. training set). Can this framework be efficiently scaled to a "many-to-many" setting for dataset-wide auditing without becoming a computational bottleneck?

2. Copyright Localization: In a scenario where a model performs "memorization" or direct patch-copying, can the resulting spatial heatmaps be used to derive a precise bounding box for the copied area? This would be highly relevant for copyright forensics.

3. Conditioning/Guidance: The derivation focuses on unconditional diffusion. How does the presence of classifier-free guidance or text conditioning affect the attribution? Does the "optimal score" assumption still hold when the density is conditioned on a text embedding $c$?

**Limitations:**

yes

**Strengths And Weaknesses:**

Strengths:
1. The nonparametric nature of the method removes the requirement for backpropagation or model retraining, making it highly suitable for "black-box" attribution of proprietary models.
2. Using the mathematical properties of the optimal score function to justify patch-level similarity is a principled approach that bridges the gap between simple heuristic similarity and complex gradient-based attribution.
3. Results on CIFAR-10 and CelebA demonstrate that the method significantly outperforms simple CLIP-based baselines and achieves parity with state-of-the-art gradient-based methods.

Weaknesses:
1. The core premise that the model's score function equals the "optimal score" assumes perfect convergence. The paper does not sufficiently address learning dynamics. In practice, a model may not reach this global optimum, and the gap between the learned score and the optimal score may introduce attribution errors that the current theory does not account for.

2. The nonparametric baselines (primarily CLIP similarity) are somewhat weak. To truly demonstrate the value of the "optimal score" derivation, the authors should compare against more sophisticated nonparametric metrics such as LPIPS (Learned Perceptual Image Patch Similarity) or DINOv2 embeddings, which are better at capturing structural and spatial relationships than global CLIP vectors.

3. The evaluation is limited to low-resolution datasets (CIFAR-10 and CelebA). While these are standard for proving "Linear Data-modeling Scores" (LDS), evaluating on a subset of a larger dataset or a more complex one (like ImageNet) would strengthen the claims.

---

> ### Author Rebuttal · Authors · 2026-03-31
>
> Thank you for your valuable review and suggestions. Below we respond to the comments in Weaknesses (W) and Questions (Q).
>
> ---
>
> ***W1: Optimal-score assumption.***
>
> Thank you for raising this concern. We agree that practical learned models do not exactly equal the optimal score. For the attribution method itself, NDA does not use the trained model at attribution time: it is inspired by the analytical patch-wise optimal-score form of Kamb & Ganguli (Eq. 9--10) and computes attribution directly from the training data, without model scores or gradients. Model training is only needed for LDS evaluation, where retrained models on random training subsets provide the model outputs used as reference in LDS computation. So the optimal-score assumption motivates the NDA design, rather than providing a learned-model quantity that NDA must compute. We will revise the discussion to make this distinction explicit. Empirically, NDA also remains effective on UViT-DDPM and UViT-Flow Matching models in Appendix D.1.
>
> ---
>
> ***W2: Stronger nonparametric baselines.***
>
> Thank you for the suggestion. We expanded the baseline set to include DINOv2 patch features:
>
> | Method | CIFAR-2 | CIFAR-10 | CelebA |
> |:--|:--:|:--:|:--:|
> | DINOv2-patch | 14.28 / 5.42 | 7.77 / 3.31 | 10.98 / 3.58 |
> | NDA (ours) | 24.88 / 15.91 | 11.81 / 7.41 | 16.89 / 13.92 |
>
> Here each entry is Val / Gen. NDA remains higher on CIFAR-2, CIFAR-10, and CelebA, for both validation and generation.
>
> ---
>
> ***W3: Larger / higher-resolution evaluation.***
>
> Thank you for pointing this out. We added ArtBench at $256\times256$:
>
> | Metric | Raw pixel | CLIP | NDA | TRAK |
> |:--|:--:|:--:|:--:|:--:|
> | Val LDS (%) | 2.58 | 8.62 | 15.67 | 12.26 |
> | Gen LDS (%) | 2.71 | 8.66 | 13.21 | 7.78 |
>
> NDA remains above Raw pixel, CLIP, and TRAK at 256×256. The corresponding multiscale study is reported in our response to R1 / W2 & Q3. This directly strengthens the claim beyond the low-resolution settings in the main paper.
>
> ---
>
> ***Q1: Many-to-many auditing.***
>
> Thank you for this question. Many-to-many auditing does not require a new formulation. As in Eq. (12), each generated image is attributed independently against the training set, so dataset-wide auditing simply repeats the same one-to-many computation over many generated images. In our implementation, Sec. 3.4 already supports batching multiple generated images. Their patches are reshaped into convolution kernels and processed in chunks, so peak memory can be controlled by the chunk size. As noted in Appendix E, the total cost grows linearly with the number of generated images, and different generated images can also be distributed across GPUs. So the main challenge for dataset-wide auditing is runtime as the number of generated images grows, not a change to the attribution rule itself.
>
>
> ---
>
> ***Q2: Copyright localization.***
>
> Thank you for this question. In principle, yes. NDA is patch-based: beyond ranking the most influential training images for a generated sample, it can also attribute the most important generated patches to specific influential training patches in the dataset. This provides a patch-level view of influence rather than only an image-level attribution, and could therefore be useful for inspecting potentially copied or highly similar regions. So if the question is whether NDA can localize influence beyond the whole-image level, our answer is yes. What we do not claim here is a complete copied-region detection pipeline: the current paper does not yet evaluate box-level localization accuracy or validate an automatic copied-region decision rule.
>
> ---
>
> ***Q3: Conditioning and guidance.***
>
> Thank you for this question. Our current score-based motivation is unconditional, so it should not be read as a formal result for text-conditioned or guidance-based sampling. This is especially important for classifier-free guidance: the guided score is typically a linear combination of unconditional and conditional scores, and therefore does not generally correspond to the exact score of a normalized density.
>
> As a limited empirical extension, we tested NDA on ArtBench using the available style labels:
>
> | ArtBench setting | Attribution pool | Val LDS (%) | Gen LDS (%) |
> |------------------|------------------|-------------|-------------|
> | Apply NDA over all training images | All training examples | 13.05 | 11.24 |
> | Restrict attribution to the query's style label | Training examples with the same style label | 15.67 | 13.21 |
>
> These results show that NDA still gives a useful attribution signal when the label is ignored, and improves further when the attribution pool is restricted to the matching style. These results show that NDA still gives a useful attribution signal when the label is ignored, and improves further when attribution is restricted to the matching style. We view this as an empirical result using available labels, not as a general result for text-conditioned or CFG-based diffusion models.

---

> > ### Author Rebuttal · Reviewer_zchd · 2026-04-04
> >
> > Thank you for addressing my concerns. I will keep my score mainly because W1 and Q3.

---

> > > ### Author Response · Authors · 2026-04-07
> > >
> > > Thank you again for your positive assessment and for the thoughtful follow-up. We also appreciate your comments on W1 and Q3, and after the rebuttal period we would clarify these two points in the manuscript to make the scope of the paper clearer.

---

### Official Review · Reviewer_1BpJ · 2026-03-12

**Soundness:** 2
**Presentation:** 4
**Significance:** 2
**Originality:** 3
**Overall Recommendation:** 4
**Confidence:** 3

**Summary:**

This paper introduces Nonparametric Diffusion Attribution (NDA), a method for attributing diffusion model generations to training data without requiring access to model parameters or gradients. The key insight is that the weighting term in the analytical optimal score function under locality and equivariance assumptions naturally encodes training example influence at the patch level. The authors reinterpret these weights as patch-wise influence scores, aggregate them across spatial locations and timesteps, and extend to multiscale representations. A convolution-based implementation avoids explicit patch unfolding for memory efficiency. Experiments on CIFAR-2, CIFAR-10, and CelebA show NDA outperforms nonparametric baselines (raw pixel, CLIP similarity) on the Linear Datamodeling Score (LDS) metric and counterfactual evaluations.

**Compliance With Llm Reviewing Policy:**

Affirmed.

**Final Justification:**

Not all of my concerns have been addressed, but the authors have justified that the strengths of this paper outweigh the weaknesses. The rebuttal has also addressed some of my concerns with baselines, hyperparameter sensitivity, and scalability.

**Key Questions For Authors:**

Most of my questions are in the fields above. A few more are:

**(Q1)**: How does NDA extend to text-conditioned diffusion models where the score function also depends on the conditioning signal? The most practically relevant attribution scenarios involve conditional models (e.g., Stable Diffusion), and the current unconditional formulation may not straightforwardly apply.

**(Q2)**: In a black-box setting, how should a practitioner choose the noise schedule parameters? Have the authors tested sensitivity to misspecified schedules (eg: using a linear schedule when the model was trained with cosine, or using the wrong number of timesteps)? This seems critical to the paper's motivating scenario

**(Q3)**: The softmax normalization in eq 11 is computed over all patches in P_Ω(S). For training sets much larger than 50K images, this denominator becomes quite expensive. Have the authors considered approximate normalization strategies, and how would approximation error affect attribution quality?

**(Q4)**: What is the variance of the LDS scores across individual test samples? Are there systematic failure modes (eg: certain image types where NDA performs particularly poorly relative to D-TRAK)?

**Limitations:**

Yes.

**Strengths And Weaknesses:**

**Strengths**:

**(S1)**: Very well presented theory and motivation. The theory is sound and the derivation of NDA is very well motivated. Reinterpreting the posterior weights from the optimal score function (Eq. 5–6) as influence scores, then extending this via the Kamb & Ganguli (2024) patch-level formulation, provides a principled justification that clearly distinguishes NDA from ad hoc similarity metrics. In general, the presentation of this paper is extremely clear.

**(S2)**: The empirical gains over nonparametric baselines are large and consistent. NDA greatly improves LDS scores compared to CLIP similarity across all three datasets, demonstrating that incorporating diffusion-process structure into a nonparametric method matters.

**(S3)**: The practical motivation is compelling. Attribution without model weight access is relevant to real-world scenarios involving proprietary models or costly training.

**(S4)**: Ablation studies are well presented and support many of the claims presented in the paper. Patch size, timestep selection, multiscale weighting, and top-k are all studied, providing clear guidance on hyperparameter choices.

**Weaknesses**:
**(W1)**: The "no model access" claim is a bit overstated. While NDA does not require model parameters or gradients, it still depends on knowledge of the forward diffusion process. NDA requires knowledge of the noise schedule parameters ⍺_t to construct noisy images x_t and to compute the properly scaled patch distances of equation 11. In a truly black-box setting (eg: attributing outputs of a proprietary API), the user would not know what noise schedule was used, how many diffusion steps were employed, or even whether the model uses a DDPM-style forward process at all (vs. flow matching, EDM, etc.). The paper's own Flow Matching experiment (Appendix D.1) uses a different interpolation scheme, yet the attribution still appears to use the DDPM noise schedule. This discrepancy deserves clarification. In practice, this means NDA requires a non-trivial assumption about the generative process,

**(W2)**: Scalability concerns are not adequately addressed. The method is demonstrated only on 32×32 and 64×64 images with at most 50K training samples. The quadratic dependence on spatial resolution concerns me about applicability to modern generative models at 256x256+ resolution. Runtime analysis (Table 6) already shows NDA is slower than D-TRAK attribution on CelebA (9:12 vs. 4:36), and this gap would likely worsen at higher resolutions. The introduction motivates the work with LAION-5B-scale settings, but no experiment approaches even moderate scale

**(W3)**: The gap to D-TRAK is understated. While the paper frames NDA as "closely matching" gradient-based methods, the gap on CIFAR-10 is ~3 points on validation and ~3.6 points on generation, and on CelebA validation it is nearly 6 points. These are not negligible. The paper could be more forthright about the settings where the method falls short and provide analysis of when and why D-TRAK's gradient information provides a meaningful advantage.

**(W4)**: Hyperparameter sensitivity is a concern. The method requires tuning patch size per timestep, γ per timestep, the timestep set, and k. The optimal patch sizes differ between datasets and between original/low-resolution scales. These choices may not transfer to new domains without LDS ground truth for tuning. To me, this somewhat undermines the claim of a practical, model-free method.

**(W5)**: Missing baselines. Given that NDA is fundamentally a patch-matching approach, comparison with established patch-based retrieval methods or modern self-supervised feature spaces (eg: DINOv2 patch features, or SSCD) would help contextualize the contribution of the diffusion-specific weighting vs performing good patch retrieval with a strong feature extractor

**(W7)**: "model-agnostic" claim. Consistency across architectures is shown only for models of similar capacity trained on the same small datasets. It could be more convincing to demonstrate consistency between models of very different quality (eg, an undertrained vs. well-converged model), or to discuss when model-agnosticism should be expected to break down. This concern is compounded by W1. The method is agnostic to the model but not to the diffusion process, which is a meaningful distinction the paper does not sufficiently draw

**(W8)**: The counterfactual evaluation is limited in scope. Removing 1,000 out of 5,000 training samples (20%) is aggressive. At such high removal rates, most reasonable attribution methods may show impact. Evaluating at smaller removal fractions (1%, 5%, 10%) where distinguishing more influential samples from merely similar ones is harder would be a more discriminative test. Additionally, only 60 test images are used, which limits statistical power.

---

> ### Author Rebuttal · Authors · 2026-03-31
>
> Thank you for your valuable review and suggestions. Below we respond to the comments in Weaknesses (W) and Questions (Q).
>
> ---
>
> ***W1, W7 & Q2: Model-access and schedule misspecification.***
>
> Thank you for pointing this out. We agree the original words should be more precise. NDA does not require the target model's parameters, gradients, architecture, or training objective, but it is not a strict black-box API method. It assumes (i) a candidate training set and (ii) a chosen diffusion forward process for attribution.
>
> In all of our experiments, including UViT-Flow Matching (**Appendix D.1**), we fix the same DDPM forward process (1000 steps, linear beta schedule) to construct $x_t$ and compute Eq. (11); we do not adapt attribution to the target model's objective or schedule. Thus the UViT-Flow result shows that NDA is not tied to the target model's training objective, not that NDA is schedule-free.
>
> On the same CIFAR-2 target model, varying only the attribution-time schedule gives LDS 24.88 (linear-1000), 23.43 (cosine-1000), 23.16 (linear-500), and 23.97 (cosine-500). These results suggest that NDA is fairly robust to different attribution-time schedules, but not schedule-free. We will therefore revise the wording from "no model access" / "model-agnostic" to the more precise "no access to model parameters or gradients" / "model-parameter-free."
>
> ---
>
> ***W2 & Q3: High-res evaluation and scalability.***
>
> We added ArtBench at $256\times256$. As noted in **R1 / W2 & Q3**, in the style-conditioned setting NDA reaches **15.67 / 13.21** (Val/Gen), above Raw pixel, CLIP, and TRAK. The CelebA stride study in **R1 / W2 & Q3** shows that stride=2 reduces runtime from 9h 12m to **1h 16m** with little LDS change and is faster than D-TRAK (4h 36m).
>
> We also tested exact patch-level oracle top-$M$ truncation on CIFAR-10 validation: for each query patch, we keep the exact top-$M$ training patches in Eq. (11), renormalize over them, and apply the same image-level aggregation as full NDA. Full NDA gives **11.81** LDS, while M=2048 and M=8192 give **11.12** and **11.34**. This suggests that exact top-$M$ truncation preserves much of the full NDA signal and motivates ANN as the next step.
>
> ---
>
> ***W3: Gap to D-TRAK.***
>
> We agree the original wording was too strong. D-TRAK is better on LDS, especially on CelebA validation (22.83 vs. 16.89). We will replace "closely matching" with the narrower claim that NDA narrows the gap to gradient-based methods while clearly outperforming nonparametric baselines. Counterfactual evidence is complementary: on the shared 100-image set, NDA gives the largest $\ell_2$ change at 1\%, 5\%, and 10\%; under CLIP similarity, NDA is best at 1\% and 10\%, while D-TRAK is best at 5\%.
>
> ---
>
> ***W4: Hyperparameter sensitivity.***
>
> Our ablations show consistent structure. On both CIFAR-2 and CelebA, smaller timesteps favor smaller patches and larger timesteps favor larger patches, though the exact optimum differs by dataset. We also find $k\approx100$, $\tau_{\max}$ saturates around 400--500, and $\gamma\approx0.75$ works well. We select these on validation and reuse them on generation; automatic selection for new domains remains open.
>
> ---
>
> ***W5: Stronger nonparametric baselines.***
>
> We added a DINOv2 patch baseline. DINOv2-patch reaches 14.28 / 5.42, 7.77 / 3.31, and 10.98 / 3.58 on CIFAR-2, CIFAR-10, and CelebA (Val / Gen), improving over patch-wise CLIP method but remaining below NDA throughout.
>
> ---
>
> ***W8: Finer counterfactual evaluation.***
>
> We reran CIFAR-2 counterfactuals on 100 images at 1\%, 5\%, and 10\% removal, a stricter follow-up to the original 20\% setting. NDA gives the largest L2 change at all three fractions: 25.974 / 26.204 / 26.759 vs. 25.582 / 25.768 / 25.987 for D-TRAK. CLIP similarity (NDA / D-TRAK) is 0.7639 / 0.7663, 0.7673 / 0.7639, and 0.7656 / 0.7664 at 1\%, 5\%, and 10\%, where NDA achieves better result at 1\% and 10\%.
>
> ---
>
> ***Q1: Extension to text-conditioned diffusion models.***
>
> Our current analysis is unconditional, so we do not claim a theorem for text-conditioned or CFG-based models. On ArtBench, applying NDA over all training images gives **13.05 / 11.24**; restricting attribution to training images with the query's style label gives **15.67 / 13.21** (Val/Gen). We present this as a simple empirical extension using available conditioning labels, not a derivation for general text conditioning or guidance.
>
> ---
>
> ***Q4: Variance across individual samples and failure modes.***
>
> We computed per-sample LDS for the 1,000 CIFAR-2 validation samples. NDA has mean **0.248**, median **0.225**, and IQR **0.228**. For 921 / 1,000 samples (92.1%), per-sample LDS is positive, meaning that NDA agrees with the ground-truth subset ranking for most samples. The class-wise mean LDS is positive for both automobile (**0.264**) and horse (**0.232**), so we do not see a strong class-specific failure mode; the harder cases appear sample-specific rather than concentrated in one class.

---

> > ### Author Rebuttal · Reviewer_1BpJ · 2026-04-03
> >
> > I thank the authors for their response. I would encourage them to ensure that the revised framing is present in the updated version of the paper.
> >
> > Some of my concerns are resolved, but I also share the concerns of reviewer X2ZT on the 256x256 resolution experiments, w.r.t the gap to D-TRAK. Also, my concern was both with resolution and dataset size, and the training set size is only 5000 images, which is still too small.

---

> > > ### Author Response · Authors · 2026-04-06
> > >
> > > Thank you for the follow-up. We agree that some of our wording in the paper was not precise, such as "closely." Sec. 4.2 already makes the main comparison clear: NDA is primarily compared with methods that do not require target-model parameters, while gradient-based methods are included as references. Table 1 is organized in the same way, with a "Without Model Access" block and a "Using Model Gradients" block. With the higher-resolution and larger-pool results included, we should describe the scalability evidence more cautiously.
> > >
> > > As resolution and pool size increase into a moderate-scale regime, NDA still gives useful attribution signal and remains clearly above simple nonparametric baselines, but performance degrades and the gap to D-TRAK remains. To address the pool-size issue more directly, we evaluated a larger 256x256 setting on ArtBench-5 using the full 12,500-image candidate pool. Applying NDA over the full pool gives 12.98 / 12.04 LDS (Val / Gen). Using the available style label to restrict the attribution pool improves this to 14.37 / 12.76. For reference, the corresponding ArtBench-5 benchmark results reported by Zheng et al. [1] are 9.79 / 8.79 for TRAK and 22.84 / 21.56 for D-TRAK.
> > >
> > > We also isolated candidate-pool size on CIFAR-10 by increasing the pool from 10K to 20K to 50K; NDA declines from 14.05 to 13.43 to 11.92, compared with 16.89 to 15.91 to 14.78 for D-TRAK and 4.94 to 4.16 to 3.47 for CLIP. Across these larger pools, NDA remains clearly above generic similarity baselines and preserves useful attribution signal, while the gap to D-TRAK remains consistent.
> > >
> > > Motivated by Q3 on Eq. (11), we also tested a simple approximation to its softmax normalization. For each query, we first rank training images with image-wise NDA, keep a small shortlist, and then compute the original patch-level NDA score only within that shortlist. In the same full-pool ArtBench-5 setting above, this raises validation LDS from 12.98 to 16.23. In the matching-style setting, it raises validation LDS from 14.37 to 17.05. This already shows that NDA is a practically meaningful method with clear promise: even a small NDA-family change yields a substantial gain in the same harder regime.
> > >
> > > Taken together, these results show that as resolution and candidate-pool size increase, NDA still gives useful attribution signal and remains clearly above Raw pixel and CLIP, while the gap to D-TRAK remains. In the 256x256 ArtBench-2 setting, NDA is also above TRAK. In other words, NDA still works meaningfully without access to target-model parameters or gradients.
> > >
> > > **Reference**
> > >
> > > [1] Zheng, Xiaosen, et al. "Intriguing properties of data attribution on diffusion models." ICLR 2024.

---

### Official Review · Reviewer_X2ZT · 2026-03-13

**Soundness:** 2
**Presentation:** 2
**Significance:** 2
**Originality:** 2
**Overall Recommendation:** 4
**Confidence:** 3

**Summary:**

The authors propose a novel, training- and gradient-free nonparametric data attribution method that leverages patch-level similarity between generated and training images by using a version of the analytical optimal score function under locality and equivariance assumptions. Convolution-based acceleration allows for multiscale representations. Experiments on the CIFAR-2, CIFAR-10, and CelebA datasets demonstrate that the proposed method performs almost as well as gradient-based approaches and outperforms existing nonparametric baselines.

**Compliance With Llm Reviewing Policy:**

Affirmed.

**Final Justification:**

The paper presents a principled, practically motivated nonparametric attribution method with a solid theoretical grounding and convincing ablations. The rebuttal adequately addressed all of my major concerns.

The remaining condition for acceptance is that the authors carry through the revisions they committed to, specifically, revise the abstract and Section 4.2 to remove "closely matching" and replace it with the honest framing described in the rebuttal, include the ArtBench-2 256×256 results in the main paper, include the Image-wise NDA ablation in the appendix and update the impact statement.

I raise my score to Weak Accept, conditional on these revisions being implemented in the camera-ready version.

**Key Questions For Authors:**

1. Could the authors precisely flesh out the novelty of their proposed approach beyond the fact that it appears to be a reinterpretation of Kamb and Ganguli (2024)? Does any part of the method require non-trivial theoretical development beyond what appears in the paper by Kamb and Ganguli (2024)?
2. Have the authors analysed how attribution quality degrades as the size of the dataset increases? Have any practical alternatives been considered in these challenging settings?
3. The quadratic complexity of the method seems to be a major issue at higher image resolutions. Can the multiscale design work well in these settings? How badly would the wall-clock time and LDS performance suffer at a resolution of $\geq 256\times256$?

**Limitations:**

Check weaknesses for specific issues that must be acknowledged in a separate limitations section.

Please update the impact statement to reflect that attribution methods have direct implications for legal proceedings, data valuation disputes, and potential misuse in false attribution claims. A brief, honest engagement is warranted, given that this paper is explicitly motivated by copyright protection and data privacy issues.

**Strengths And Weaknesses:**

**Strengths:**
1. The problem statement is well motivated for the practical setting where gradients may not be available, and training is expensive or impossible.
2. Deriving the attribution score based on the optimal score function provides a neat, clear theoretical grounding.
3. Experiments provide good ablations on patch size, timestep and $\gamma$, and the proposed method shows consistent improvements over existing baselines.

**Weaknesses:**
1. The core technical idea seems to be an incremental extension of the equivariant local score formulation from Kamb and Ganguli (2024) with a clever choice of the weighting function. It appears that the novelty lies in the reinterpretation and aggregation of existing results, which is non-trivial but insufficient. The authors should clearly mention their specific contribution.
2. Experiments can be further strengthened by reporting results on higher resolution images, e.g., FFHQ-256 or ImageNet-64. The quadratic complexity would be an issue at those scales, and it would be interesting to see how the method fares in those settings.
3. The claim in the abstract about closely matching the performance of gradient-based methods must be softened since the gap in performance is about 25% in the CelebA validation case.
4. The patchwise influence score in Eq. (11) requires computing softmax over all patches in the training set, and this will get expensive as datasets get large. How does the attribution quality degrade as the number of samples increases?
5. The claim that the attribution scores are consistent across a variety of architectures and training regimes must be softened unless sufficient experimental evidence is provided to back it up. Currently, experiments are limited to the UNet architecture on small datasets.

---

> ### Author Rebuttal · Authors · 2026-03-31
>
> Thank you for your valuable review and suggestions. Below we respond to the comments in Weaknesses (W) and Questions (Q).
>
> ***W1: Novelty relative to Kamb & Ganguli (2024).***
>
> Thank you for raising this concern. Our novelty is methodological: Kamb \& Ganguli (2024) characterize the local score form, but do not define a data attribution method. NDA turns that structure into a concrete attribution pipeline: a patch-wise influence score (Eq. 11), image-level aggregation (Eq. 12), a multiscale extension (Eqs. 13--15), and the convolution-based implementation in Sec. 3.4. We also added controls showing that the gain is not from patchification alone. Image-wise NDA is much weaker than NDA (CIFAR-2 / CIFAR-10 / CelebA val: 9.96 / 6.07 / 5.82 vs. 24.88 / 11.81 / 16.89). Replacing NDA with simpler patch scores is also much weaker: patch L2 gives 5.94 / 1.60 / 7.81 and noise-weighted patch L2 gives 5.34 / 1.53 / 7.43; patch-wise Raw pixel and patch-wise CLIP are likewise far below NDA. These results are reported in App. D.2, Tables 3--5. Thus the gain is not from patchification alone, but from the score-based attribution design.
>
> ---
>
> ***W2 & Q3: Higher-resolution scalability, multiscale design, and wall-clock cost.***
>
> Thank you for raising this concern. We added the experiment of ArtBench dataset at $256\times256$ resolution:
>
> | Metric | Raw pixel (cosine) | CLIP similarity (cosine) | NDA | TRAK | D-TRAK |
> |:--|:--:|:--:|:--:|:--:|:--:|
> | Val LDS (%) | 2.58 | 8.62 | **15.67** | 12.26 | 27.61 |
> | Gen LDS (%) | 2.71 | 8.66 | **13.21** | 7.78 | 24.16 |
>
> We also ablated original-resolution, low-resolution, and multiscale variants at $256\times256$ on Artbench validation dataset:
>
> | Setting | t=100 | t=200 | t=300 | t=400 |
> |---------|-------|-------|-------|-------|
> | Original resolution | 13.81 | 13.21 | 12.34 | 11.97 |
> | Low-resolution | 14.22 | 13.96 | 12.67 | 12.46 |
> | Multiscale | 14.96 | 14.53 | 13.21 | 12.88 |
>
> The multiscale variant is best at all four tested timesteps, yielding 15.67 after aggregation.
>
> For wall-clock cost, we report a CelebA stride study using spatial patch subsampling during patch extraction:
>
> | Setting | Val LDS (%) | Gen LDS (%) | Runtime |
> |---------|-------------|-------------|---------|
> | NDA stride=1 | 16.89 | 13.92 | 9h 12m |
> | NDA stride=2 | 17.20 | 13.73 | 1h 16m |
> | D-TRAK | 22.83 | 16.84 | 4h 36m |
>
> On CelebA, stride=2 cuts runtime from 9h 12m to 1h 16m with little LDS change (16.89 / 13.92 $\rightarrow$ 17.20 / 13.73 on Val / Gen), so we use stride=2 for ArtBench.
>
> ---
>
> ***W3: "Closely matching" claim.***
>
> Thank you for pointing this out. We agree the original wording is too strong. D-TRAK remains better on LDS in several settings, especially CelebA validation (22.83 vs. 16.89). We will replace "closely matching" with the narrower claim that NDA narrows the gap to gradient-based methods while clearly outperforming nonparametric baselines. We do not claim parity with D-TRAK on LDS.
>
> ---
>
> ***W4 & Q2: Softmax cost and attribution quality at scale.***
>
> Thank you for raising this concern. We ran a controlled CIFAR-10 scaling study by sampling class-balanced 10K/20K/50K training pools (1K/class, 2K/class, full set), with all other settings fixed; for each pool, LDS is computed using $M=16$ random 50%-subsets with 3 seeds to keep the computational cost tractable.
>
> | $N$ | NDA | D-TRAK | CLIP (cosine) |
> |---:|:---:|:---:|:---:|
> | 10K | 14.05 | 16.89 | 4.94 |
> | 20K | 13.43 | 15.91 | 4.16 |
> | 50K | 11.92 | 14.78 | 3.47 |
>
> From 10K$\rightarrow$50K, all three methods decrease as $N$ increases. NDA does not drop faster than D-TRAK and stays well above CLIP at every tested scale.
>
> ---
>
> ***W5: Consistency across architectures and training objectives.***
>
> Thank you for raising this concern. We agree the original claim was too broad and should be tied to the settings we actually tested. In the main paper, all results use U-Net target models trained with DDPM. **Appendix D.1** evaluates NDA on target models trained with different architectures and objectives: UViT-DDPM and UViT-Flow Matching on CIFAR-2, CIFAR-10, and CelebA. Across these additional settings, NDA consistently outperforms Raw pixel and CLIP and remains close to D-TRAK; for example, on UViT-DDPM / CIFAR-10, NDA is 6.07 versus 6.62 for D-TRAK, and on UViT-Flow Matching / CIFAR-2, NDA is 15.61 versus 15.20. We therefore revise the claim to the narrower statement that NDA remains effective across the tested target-model settings, namely U-Net/DDPM in the main paper and UViT/DDPM plus UViT/Flow-Matching in **Appendix D.1**, rather than claiming general consistency across architectures and training regimes. We will also clarify that NDA requires a candidate training set, that higher-resolution scaling still involves a cost-quality tradeoff, and that attribution scores should not be treated as definitive legal evidence without additional validation.

---

> > ### Author Rebuttal · Reviewer_X2ZT · 2026-04-03
> >
> > Thank you for your detailed rebuttal. I appreciate the effort that the authors have put in to meaningfully address most of my concerns.
> >
> > However, one concern is not fully resolved and I believe that the manuscript can be improved by clarifying this. The ArtBench $256\times 256$ results provide a more complete picture and the stride trick is a practical contribution at higher resolutions. Validation LDS of NDA is reported as 15.67% which has a 43% relative gap compared to Validation LDS of D-TRAK reported as 27.61% which is substantially larger than the 26% relative gap observed for CelebA $64\times 64$. This seems to indicate that at higher resolutions NDA _widens_, not narrows, the gap to gradient-based methods.
> >
> > __How should the paper's framing be updated to honestly characterize NDA's relative performance with increasing resolution, given that this is the setting most relevant to the paper's stated motivation of attribution in large-scale or proprietary generative model settings?__
> >
> > I believe that the authors should attempt to offer a principled explanation for why the widening gap does not undermine the paper's contribution in realistic settings.

---

> > > ### Author Response · Authors · 2026-04-06
> > >
> > > Thank you for the follow-up. We agree that some of our wording in the paper was not precise, such as "closely." Sec. 4.2 already makes the main comparison clear: NDA is primarily compared with representative methods that do not require access to model parameters, while gradient-based methods are included as references. Table 1 is organized in the same way, with a "Without Model Access" block and a "Using Model Gradients" block. With the 256x256 result included, we should describe this comparison more carefully.
> > >
> > > On ArtBench-2 (256x256), NDA reaches 15.67 / 13.21 LDS (Val / Gen), compared with 12.26 / 7.78 for TRAK and 27.61 / 24.16 for D-TRAK. So in this high-resolution setting, NDA remains above TRAK but below D-TRAK, and the gap to D-TRAK is larger than in our lower-resolution experiments. A fairer summary is that, among the methods we evaluate without target-model parameters, NDA performs the best: it clearly outperforms the other nonparametric baselines and is the closest of that group to the gradient-based references, while D-TRAK remains better in this evaluated high-resolution setting.
> > >
> > > We also do not want to read CelebA -> ArtBench as a pure resolution-only scaling curve, since the dataset, target model, and benchmark protocol also change. The safe conclusion from the current evidence is simply that, in the tested 256x256 regime, the gap to D-TRAK is larger.
> > >
> > > We think this does not contradict the paper's contribution. NDA is a practical nonparametric data attribution method that uses the candidate training set together with an attribution-time forward process, without access to model gradients. D-TRAK, in contrast, uses gradients of the trained target model, which carry model-specific information about how training examples influence the generation of the output. In harder 256x256 settings, patch matching from data alone becomes less reliable, while D-TRAK can still use that additional model-specific information. We think the larger gap to D-TRAK is consistent with the setting NDA is designed for: NDA does not use target-model gradients, while D-TRAK does. The contribution we intend to claim is that NDA provides a practical nonparametric attribution method based on the local optimal-score weighting, with patch-based spatial interpretability and lighter resource requirements in restricted-access settings. In such settings, it can already provide useful attribution evidence even when target-model gradients are unavailable.

---

### Decision · Program_Chairs · 2026-04-30

**Decision:**

Accept (regular)

**Comment:**

This paper proposes a novel data attribution method for diffusion models, which defines the attribution score as the similarity between the patches in a generated samples to those in the training data. The similarity is measured based on the analytical optimal score function under locality and equivariance assumptions. A key feature of the proposed method is non-parametric as it does not require access to the model parameters. Empirical results suggest that the proposed method achieves competitive performance as gradient-based data attribution methods in terms of the LDS metric.

Reviewers generally agreed that this is a well-written paper with a novel and principled data attribution method, and that the problem setup where gradients of models may not be available is practical. However, there are debates about the "non-parametric" property is an advantage or disadvantage. While not requiring access to model gradients could be helpful for certain scenarios, the proposed method could be viewed as less about interpreting a particular model but more about measuring the similarity between the data. The fact that it could be applied to models other than diffusion models also undermines the paper title suggesting the proposed method is designed for diffusion models.